# Eukaryotic Pif1 helicase unwinds G-quadruplex and dsDNA using a conserved wedge

Zebin Hong[1,3], Alicia K. Byrd ●[2,3] ✉, Jun Gao[2], Poulomi Das[1], Vanessa Qianmin Tan[1], Emory G. Malone[2], Bertha Osei[2], John C. Marecki ●[2], Reine U. Protacio ●[2], Wayne P. Wahls ●[2], Kevin D. Raney ●[2] ✉ & Haiwei Song ●[1] ✉

G-quadruplexes (G4s) formed by guanine-rich nucleic acids induce genome instability through impeding DNA replication fork progression. G4s are stable DNA structures, the unfolding of which require the functions of DNA helicases. Pif1 helicase binds preferentially to G4 DNA and plays multiple roles in maintaining genome stability, but the mechanism by which Pif1 unfolds G4s is poorly understood. Here we report the co-crystal structure of *Saccharomyces cerevisia*e Pif1 (ScPif1) bound to a G4 DNA with a 5′ single-stranded DNA (ssDNA) segment. Unlike the *Thermus oshimai* Pif1-G4 structure, in which the 1B and 2B domains confer G4 recognition, ScPif1 recognizes G4 mainly through the wedge region in the 1A domain that contacts the 5′ most G-tetrad directly. A conserved Arg residue in the wedge is required for Okazaki fragment processing but not for mitochondrial function or for suppression of gross chromosomal rearrangements. Multiple substitutions at this position have similar effects on resolution of DNA duplexes and G4s, suggesting that ScPif1 may use the same wedge to unwind G4 and dsDNA. Our results reveal the mechanism governing dsDNA unwinding and G4 unfolding by ScPif1 helicase that can potentially be generalized to other eukaryotic Pif1 helicases and beyond.

Guanine-rich DNAs can fold into noncanonical DNA structures called G-quadruplexes (G4s). G4s are formed by the stacking of several G-tetrads, each of which consists of four guanine bases held together by Hoogsteen hydrogen bonding[1]. In the human genome, G4s are enriched in the promoter regions[2], telomeric DNA ends[3], meiotic and mitotic double-strand break (DSB) hotspots[4], mitochondrial DNA deletion breakpoints[5], ribosomal DNA (rDNA)[6] and untranslated regions[7]. The highly stable G4s can impede many cellular processes such as replication[8], transcription[9], and translation[7], and the failure to resolve G4s results in genome instability. Cells

have evolved many kinds of helicases that can unfold G4 structures, including DHX36[10], Pif1[8], XPD[11], and members of the RecQ helicase family (RecQ, WRN, and BLM)[12]. Among these, DHX36 and Pif1 are two helicases that have been studied extensively for understanding G4 unfolding and structures of the complexes with G4 DNA have been solved. The co-crystal structure of a truncated form of bovine DHX36 bound to the parallel G4 from the c-Myc promoter showed that recognition of G4 is conferred by the N-terminal DHX36-specific motif (DSM) together with the OB-fold like subdomain[13]. Structural comparison together with the single-molecule FRET

[1]Institute of Molecular and Cell Biology (IMCB), Agency for Science, Technology and Research (A*STAR), Proteos, Singapore, Republic of Singapore. [2]Department of Biochemistry and Molecular Biology, University of Arkansas for Medical Sciences, Little Rock, AR, USA. [3]These authors contributed equally: Zebin Hong, Alicia K. Byrd. ✉e-mail: AKByrd@uams.edu; raneykevind@uams.edu; haiwei@imcb.a-star.edu.sg

analysis suggests a model wherein G4 binding alone induces conformational changes of the helicase core, thereby driving G4 unfolding one nucleotide at a time[13].

Pif1 helicase plays multiple roles in maintaining genome stability in both the nucleus and mitochondria. In the nucleus, Pif1 inhibits telomerase at both telomeres and DSBs[14–17], processes Okazaki fragments[18,19], promotes break-induced replication[20], regulates rDNA replication[21], and prevents replication pausing at G-quadruplex structures[8,22,23], tRNA genes[24,25], and R-loops[26]. We and others have solved several crystal structures of Pif1 in complex with either a ssDNA or a forked dsDNA including *Bacteroides sp* (BaPif1)[27,28], ScPif1[29], and human Pif1[27,30] but how Pif1 unfolds G4 remains elusive. Recently, the crystal structure of *Thermus Oshimai* Pif1 (ToPif1) bound to a G4 DNA with a short ssDNA segment on both sides of the G-tetrad has been reported[31]. The structure shows that two ToPif1 molecules bind to 5′ and 3′ tails to form a dumbbell-shaped structure with the molecule bound to the 5′ ends seemingly being in an active unwinding state. Surprisingly, ToPif1 recognizes G4 mainly through the poorly conserved 2B domain, and although the previously identified wedge region is critical for separating the incoming strands of duplex DNA[27,32,33], it has no contact with the G4. Therefore, this structure provided very little information on the mechanism by which ToPif1 unfolds G4.

To reveal the structural basis for unfolding G4 by Pif1, we determined the structure of ScPif1 in complex with a G4 DNA containing a 5′ ssDNA segment. The structure showed that the wedge region contacts the ssDNA/G4 junction, poised for unfolding G4. Predictions about molecular mechanisms from the structural data were tested directly by determining the effects of single amino acid substitutions on reaction kinetics of Pif1 in vitro and on its biological activities in vivo. This combined, systematic approach revealed that ScPif1 used a conserved wedge to unwind dsDNA and unfold G4 DNA.

## Results
### Structural overview
Guanine-rich oligonucleotides can fold into G4s but tend to adopt a mixture of multiple G4 conformations in solution, thereby hindering structural studies. Previous studies showed that ScPif1 binds to a parallel G4 DNA tightly but unfolds it slowly compared to an antiparallel G4[34]. To search for a G4 substrate suitable for structural determination, we used 1D NMR to identify AT11 (Supplementary Fig. 1a and Supplementary Table 1), an anti-proliferative DNA sequence that inhibits cell growth[35] that binds to a truncated ScPif1 (residues 236–753, referred as ScPif1 thereafter, Fig. 1a) in solution[35]. G4 structures in the cell are likely have ssDNA tails on both the 5′- and 3′-end because G4 structures are more likely to form in ssDNA than dsDNA and G4 formation destabilizes nearby duplexes[36]. This, combined with ScPif1's 5′−3′ directionality led us to study the interaction of ScPif1 with a 5′-ssDNA tailed G4. The CD spectrum of the complex of ScPif1 with T7-AT11 (Supplementary Table 1) demonstrated characteristics of parallel G-quadruplex formation similar to its free form (Supplementary Fig. 1b), with the characteristic maximum at 260 nm indicative of parallel G4 formation in the presence and absence of ScPif1. This confirms that the structure of AT11 remained intact on complex formation with ScPif1. A fluorescence anisotropy binding assay showed that ScPif1 binds to AT11 in the absence (AT11-FAM) or the presence of a 5′ tail (T7-AT11-FAM) with high affinities (Supplementary Fig. 1c and Supplementary Table 1). We then went on to crystallize ScPif1 in complex with ADP·AlF4− and T7-AT11 containing a 5′ T7 ssDNA loading site (designated as ScPif1-G4) and determined its structure at a resolution of 3.5 Å (Fig. 1b). The asymmetric unit of the crystal structure contains two essentially identical ScPif1-G4 complexes. For simplicity, we only describe one ScPif1-G4 complex (chain C for ScPif1 and chain D for G4 DNA). The structure of ScPif1-G4 showed that the 5′ssDNA tail of G4 binds to the ScPif1 similar as observed in the structure of ScPif1 complexed with a ssDNA[29] (Fig. 1b). The AT11 portion of the DNA

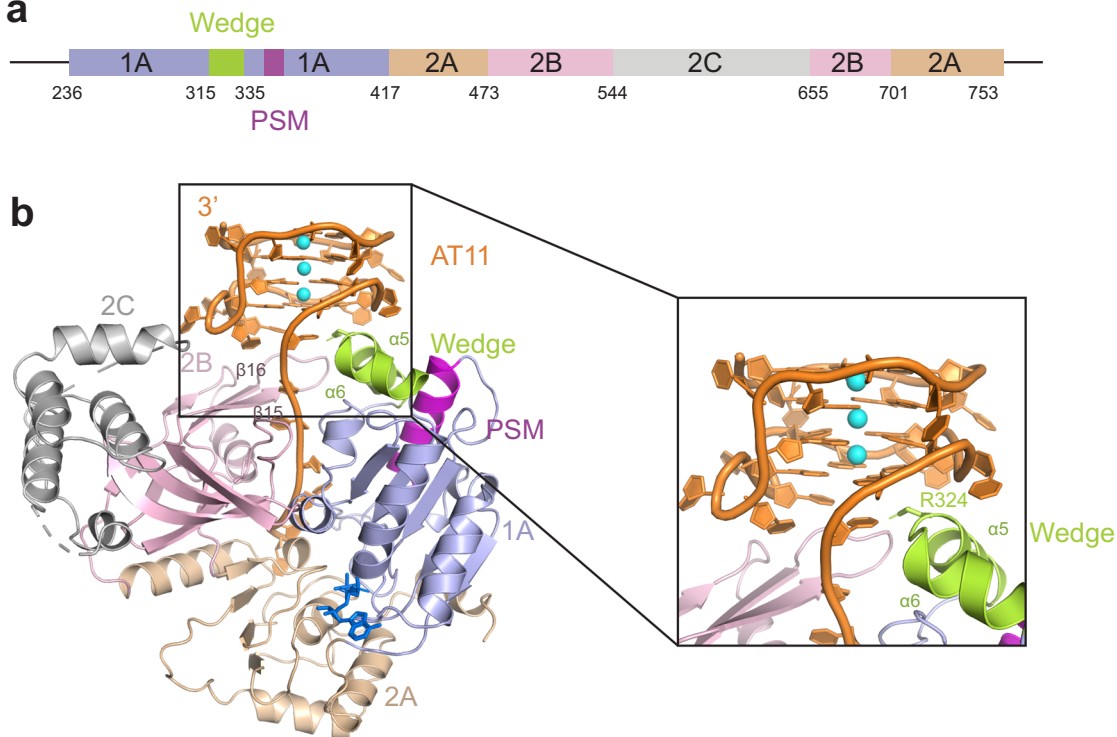

**Fig. 1 | Overall structure of ScPif1 complexed with G-quadruplex DNA.**
**a** Schematic of the truncated ScPif1 used in this study. **b** Cartoon representation of the crystal structure of ScPif1-G4 with the conserved domains and motifs color-coded as in (**a**) (1A in light blue, 2A in wheat, 2B in light pink, 2C in gray, wedge in limon, and Pif1 signature motif (PSM) in magenta). The AT11 G4 DNA is colored in orange and the ADP·AlF4− and K+ ions are shown as blue sticks and cyan spheres, respectively. The key residue R324 involved in the interaction is shown as sticks.

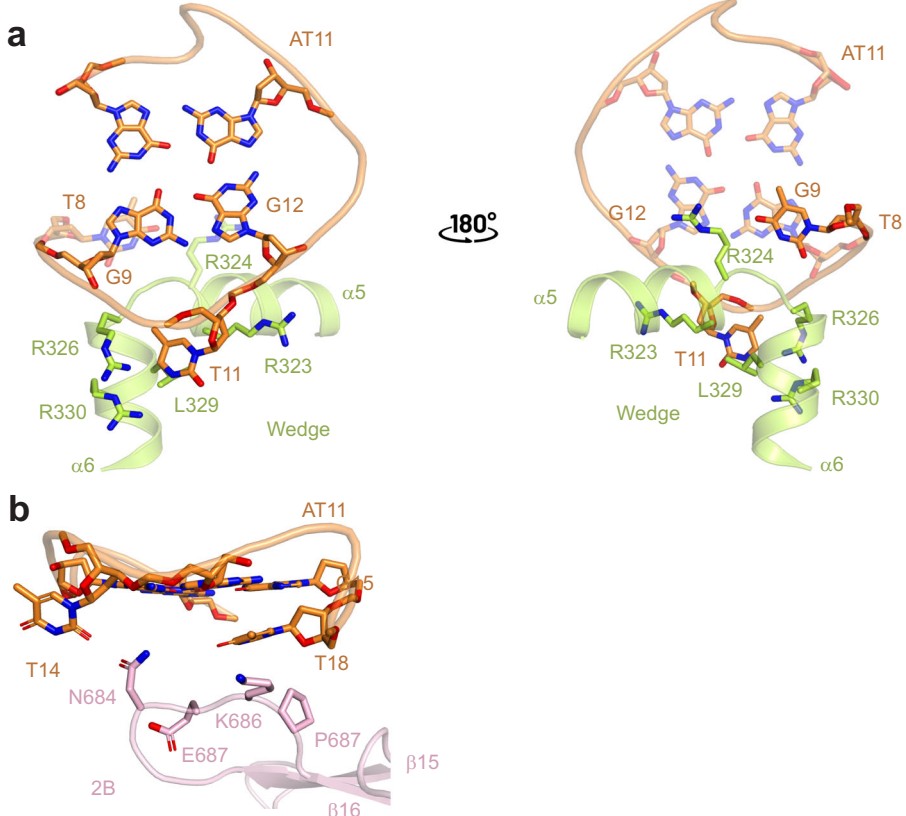

**Fig. 2 | Interactions of ScPif1 with G4 DNA. a** Major interaction between the wedge and AT11 G4 DNA. The wedge contacts the ssDNA-G4 junction and the 5′ most G-tetrad with the α5-α6 loop contributing the majority of the interactions with G4 DNA. **b** Minor interaction between 2B domain and AT11 G4 DNA. The β15-β16 loop in the 2B domain interacts solely with the peripheral region of the G4 DNA and has no contact with the G-tetrad. The residues involved in the interaction are shown as sticks and color-coded as in Fig. 1.

substrate is well ordered in the electron density map (Supplementary Fig. 2a) and adopts a G4 conformation very similar to that of AT11 alone, indicating that ScPif1 did not disrupt the G4 structure (Supplementary Fig. 2b). Similarly, the G4 structure was also intact in the ToPif1-G4 structure[31]. In support of these observations, the binding of ScPif1 to the G4 DNA does not affect the G4 structure as determined by DMS foot printing[34]. In the structure of ScPif1-G4, the conformation of ScPif1 is very similar to that of ScPif1 bound to a ssDNA[29] except for the 2B domain, which rotates 9° upon G4 binding relative to ssDNA bound ScPif1 (Supplementary Fig. 3). Similar 2B domain rotation was also reported in ToPif1 bound to a G4 DNA[31].

### Interaction of ScPif1 with G4 DNA

As shown in Fig. 1b, ScPif1 recognizes G4 mainly through the wedge region containing α5 and α6 with additional interactions contributed by the β15-β16 loop in the 2B domain. The interaction between ScPif1 and G4 buries a total solvent-accessible surface area of 528 Å². In the wedge region, the loop connecting α5 and α6 contributes the majority of the interactions with G4. The prominent feature of the ScPif1-G4 interactions is that R324 interacts with G12 in the 5′ most G-tetrad (G9.G12.G15.G19) through cation-π interaction and with T8 (the first nucleotide preceding the 5′ most G-tetrad) through van der Waals contacts, which in turn stacks against G9 (Fig. 2a). Additional interactions involve R326 which contacts the phosphate backbone between G9 and G10 and the base of T11, and R323, which interacts with the phosphate backbone connecting T11 and G12, and L329 and R330, which interact with the base of T11 through van der Waals contacts. Consistent with the critical role of R324 and R326 in G4 binding, mutation of R324 and R326 significantly reduced the G4 unfolding activities of ScPif1[29]. To further examine the functional role of R324, we

expressed and purified R324 single mutants R324A, R324E, R324W, and R324Y. These R324 variants showed similar CD spectra to that of wild type ScPif1, indicating that these single mutations do not disrupt protein folding (Supplementary Fig. 1d). Furthermore, our binding assays showed that the R324E mutant exhibited substantially reduced binding to AT11 G4 DNA (Supplementary Fig. 1e).

Compared with the wedge region, the β15-β16 loop in the 2B domain interacts solely with the peripheral region of the G4 and has no contact with the G-tetrad (Fig. 2b). Specifically, N684 is hydrogen-bonded with the base of T14 while K686 and P687 make van der Waals contacts with the base of T18, which in turn stacks against the base of G15 in the 5′ most G-tetrad. These observations strikingly differ from those reported in the structure of ToPif1-G4 complex wherein the corresponding β hairpin in ToPif1 directly interacts with the 5′ most G-tetrad[31]. Because the 2B domain is not well conserved across Pif1 family members (Supplementary Fig. 4) the mode of G4 recognition by ToPif1 does not appear to be conserved in other Pif1 family enzymes.

Previously, we showed that the Pif1 signature motif in BaPif1 stabilized the conformation of regions involved in ssDNA binding[27]. In the structure of ScPif1-G4, the Pif1 signature motif not only contacts the regions involved in ssDNA binding but also interacts with the two α-helices in the wedge region predominantly through extensive hydrophobic contacts, thereby stabilizing the wedge region (Supplementary Fig. 5). The wedge is critical for separating the incoming strands of duplex DNA so the stabilization of the wedge by the Pif1 signature motif would maintain its rigidity for exerting its helicase activity. In support of the critical role of Pif1 signature motif in stabilizing the wedge, substitution of the first 15 residues (Asp352 to Gln366) in this motif with 15 Ala residues abolished the multiple functions of ScPif1 in vivo and its ATPase activity[37]. Further supporting

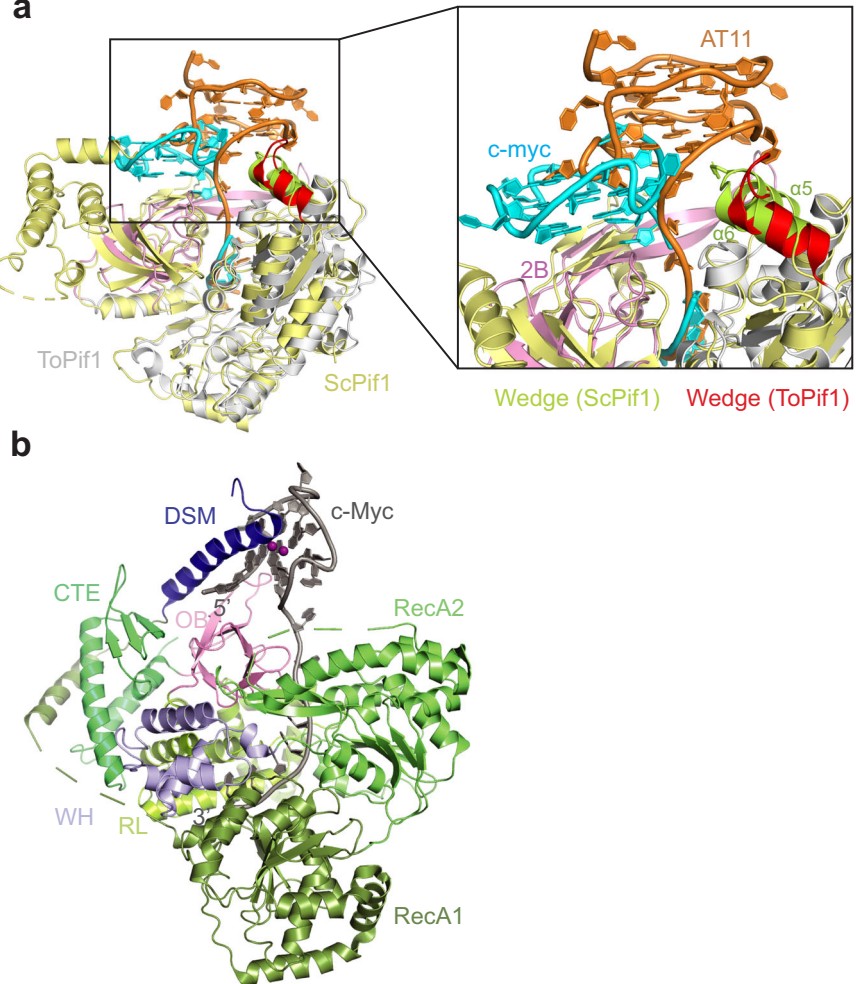

**Fig. 3 | Structural comparison with ToPif1-G4 and DHX36-G4. a** Structural superposition of ScPif1-G4 with ToPif1-G4 highlighting the conformational difference of bound G4 DNAs. The cartoon representations show ScPif1 (yellow) and ToPif1 (gray) (PDB code: 7OAR), with their wedge regions colored in lemon and red, respectively. The 2B domain of ToPif1 is colored in light pink while AT11 and c-Myc are colored in orange and cyan, respectively. **b** Cartoon representation of the DHX36-G4 structure (PDB code: 5VHE) with the DSM colored in blue, RecA1 in dark green, RecA2 in green, WH in violet, RL in lemon, OB domain in pink, CTE in limegreen and c-Myc G4 DNA in dark gray.

the importance of the Pif1 signature motif in supporting the wedge, a mutation resulting in a L319P substitution in the signature motif of human Pif1 is associated with a predisposition to breast cancer, and the corresponding L430P mutation in *S. pombe* Pfh1 lacks function in both the nucleus and mitochondria[38]. The equivalent I118P mutation in BaPif1 severely impairs ssDNA binding, dsDNA unwinding, and G4 unfolding, suggesting that the Pif1 signature motif plays an important role in Pif1 activity although its role is indirect as it does not contact the DNA itself but instead interacts with the wedge region[27].

**Structural comparison with ToPif1-G4 and DHX36-G4**
Structural superposition of ScPif-G4 with ToPif1-G4 showed that the two Pif1 molecules adopted very similar conformations upon G4 binding with α5 and α6 in the wedge region of ScPif1 corresponding to the extended region and α5 in the wedge region of ToPif1, respectively (Fig. 3a). The striking difference between ScPif-G4 and ToPif1-G4 is the orientation of the parallel G4. In ScPif-G4, the 5′ most G-tetrad interacts mainly with the wedge region whereas the 5′ most G-tetrad contacts predominantly the 2B domain of ToPif1[31] (Fig. 3a). As such, the G4 in ScPif1 rotates ~180° with respect to the G4 in ToPif1. To gain greater insight into how helicases recognize and act upon G4 structures, we then compared the structure of ScPif1-G4 to that of the helicase DHX36, which also binds to and unwinds G4 DNA[13]. The crystal

structure of bovine DHX36-G4 (Fig. 3b) showed that DHX36 contacts the 5′ most G-tetrad of the bound G4 mainly through its DSM[13]. Since DHX36 is a 3′–5′ G4 resolving DNA helicase, the interaction of DSM with G4 suggests that DSM is crucial for specific recognition of G4 but not directly involved in G4 unfolding[13]. This is dramatically different from the wedge region in ScPif1 wherein it is important for both G4 interaction and unfolding (Fig. 3).

**The wedge region is a conserved structural feature for unwinding G4 and dsDNA**
Several Pif1 structures have been determined which all contain a wedge region positioned to interact with the incoming DNA[27–30]. These structures allowed us to examine whether the wedge region is conserved structurally across Pif1 family members. As shown in Fig. 4a, the wedge region in prokaryotic Pif1 helicases comprises an extended region followed by an α-helix while the wedge in eukaryotic Pif1 helicases consists of two α-helices with a short loop in between. Close examination of the wedge regions showed that R324 in ScPif1, which is critical for G4 recognition, is conserved in Pif1 from humans (R290) and *Candida albicans* (R455) while R324 is substituted by K87 in BaPif1 (Fig. 4b and Supplementary Fig. 4). No residue corresponding to R324 in ScPif1 can be identified in Pif1 from *Thermus oshimai* or *Deferribacter desulfuricans* (Supplementary Fig. 4).

## a

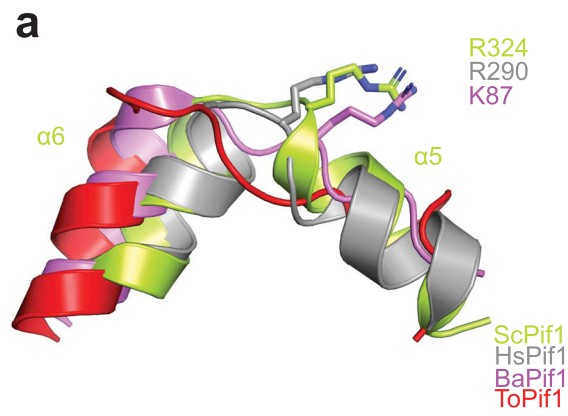

## b

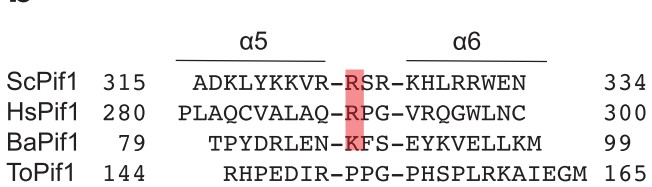

**Fig. 4 | Structural alignment of the wedge of Pif1 orthologs. a** Structural alignment of the wedge regions of Pif1 from *Saccharomyces cerevisiae*, *Homo sapiens*, *Bacteroides sp.*, and *Thermus oshimai*. The cartoon representations show the superposed wedge regions of ScPif1 (limon), HsPif1(gray) (PDB code: 5FHH), BaPif1(purple) (PDB code: 5FHE) and ToPif1(red) (PDB code: 7OAR). The wedge region in prokaryotic Pif1 helicases has an extended region followed by an α-helix while the wedge in eukaryotic Pif1 helicases contains two α-helices connected by a short loop. The key positively charged residues in the wedge regions, R324 in ScPif1, R290 in HsPif1, and K87 in BaPif1 are shown in sticks. **b** Structure-based sequence alignment of the wedge regions of Pif1 orthologs.

Previous studies showed that a pin-like β hairpin in other SF1B helicases RecD2 and Dda functions as a wedge for dsDNA unwinding[39,40]. Structural comparison of ScPif1-G4 with RecD2-ssDNA showed that the pin in RecD2 and the wedge in ScPif1 are located in similar positions and are appropriately positioned to separate incoming strands of dsDNA (Supplementary Fig 6). Given that ScPif1 has both dsDNA unwinding and G4 unfolding activities, this structural observation suggests that Pif1 may use the same wedge to unwind dsDNA and unfold G4 DNA.

### The effect of the wedge mutations on dsDNA unwinding and G4 DNA unfolding

Because R324 is located in the wedge and directly contacts the 5′ most G-tetrad, we tested the importance of interactions at this site for melting duplex and G4 structures. As reported previously, ScPif1 preferentially unwinds a forked duplex (Fig. 5a, c and Supplementary Fig. 7a, b)[41,42]. For both forked and non-forked duplexes, ScPif1 variants that are likely to retain the ability to interact with the G-tetrad due to the ability of aromatic residues to stack with a DNA base (R324W and R324Y) retain their helicase activity[13,27,39,43] (Fig. 5b, c and Supplementary Fig. 7a, b). In fact, R324W and R324Y ScPif1, which would be predicted to stack with G12 in the 5′ most G-tetrad produced more product in a single-turnover reaction than wild type ScPif1. On the other hand, R324A and R324E interfere with interactions of ScPif1 with the 5′ most G-tetrad, and these variants have minimal duplex unwinding activity in a single-turnover reaction.

R324A and R324E ScPif1 also exhibit reduced activity for unfolding the AT11 G4 (Fig. 5d, e and Supplementary Fig. 7c) and the c-Myc G4 (Fig. 5f and Supplementary Fig. 7d). The reduction in helicase activity of R324A and R324E cannot be explained by differences in ATP

hydrolysis as they hydrolyze ATP at a similar rate to WT ScPif1 and at a faster rate than R324W and R324Y ScPif1 (Supplementary Fig. 7e). However, unlike with dsDNA, R324W and R324Y ScPif1 also exhibit slightly reduced G4 DNA unfolding activity. Because the trapped product of the G4 DNA unfolding assay (Fig. 5d) is a non-forked 5′-overhang duplex and R324W and R324Y Pif1 unwound a similar substrate better than WT Pif1, it is not clear whether the reduced rate of product formation of R324W and R324Y relative to WT Pif1 is due to a reduction in the G4 DNA unfolding rate or a competition between G4 DNA unfolding and unwinding the dsDNA product. To address this possibility, we measured G4 DNA unfolding activity using a G4 reporter assay. Previous studies have shown that unwinding of a short duplex 3′ to the G4 DNA (Supplementary Fig. 8a) serves as a reporter for G4 DNA unfolding by Pif1[34,44]. Importantly, the products of this reaction are ssDNA and a 3′-overhang duplex, neither of which is a Pif1 substrate. Unfolding of a c-Myc reporter substrate by WT, R324W, and R324Y ScPif1 occurred at similar rates, suggesting that the reduced rate of product formation by R324W and R324Y ScPif1 in the G4 DNA assay in Fig. 5 is due to unwinding of the product, not differences in G4 DNA unfolding rates. Thus, for both duplex substrates and both G4 substrates, R324W and R324Y function similarly to WT ScPif1 while R324A and R324E have reduced activity, indicating that interaction of the residue at 324 with the incoming DNA is important for both dsDNA unwinding and G4 DNA unfolding.

### The effect of the wedge mutations on Pif1 activities in vivo

Because the R324 variants have reduced enzymatic activity but still retain some helicase activity, we tested the ability of the R324 variants to perform Pif1 functions in vivo. In yeast, ScPif1 promotes formation of long flaps that require processing by both Dna2 and FEN1 nucleases during Okazaki fragment maturation[45]. Deletion of Dna2 is lethal because of the failure to complete processing of these Okazaki fragments and can be rescued by deletion of Pif1[37]. We tested the function of the R324 variants in Okazaki fragment processing by transforming plasmids expressing Pif1 variants into *pif1Δ dna2Δ* haploid yeast (Fig. 6a, b). Cells with a functional Pif1, including WT, R324W, R324Y, and R324K, were not viable. Although expressed at similar levels (Supplementary Fig. 9), cells expressing no Pif1 (empty vector), ATPase deficient Pif1 (K264A), R324A, R324E, and R324N were viable, indicating that *dna2Δ* lethality was suppressed and these variants are not functional for Okazaki fragment processing. Thus, the ScPif1 variants in which the interaction with the incoming DNA has been reduced are not functional for Okazaki fragment processing. However, the variants with a residue at 324 that can interact with the DNA are functional for Okazaki fragment processing, indicating that they are able to extend the 5′-flaps on Okazaki fragments such that cleavage by Dna2 is required. Interestingly, substitution of R324 with either another positively charged residue (R324K) or an aromatic residue (R324W or R324Y) resulted in an enzyme that could extend the 5′-flaps on Okazaki fragments, suggesting that either electrostatic or stacking interactions with the DNA by the residue at position 324 support ScPif1 function.

Yeast lacking Pif1 has severe mitochondrial deficiencies and is unable to use glycerol, a non-fermentable carbon source, as the sole carbon source[37]. Plasmids encoding Pif1 variants were transformed into a heterozygous diploid strain (YCG59, *PIF1/pif1::NatMX6, DNA2/dna2::KanMX6*). After tetrad dissection, haploid cells carrying plasmids were grown on selective media containing either glucose or glycerol. Cells lacking Pif1 (Empty vector) or expressing ATPase deficient Pif1 (K264A) were unable to grow on glycerol, but all cells expressing R324 variants were viable on glycerol (Fig. 6c). This surprising result indicates that interactions of the residue at position 324 with the incoming DNA are not required for Pif1 to support respiration in the mitochondria.

ScPif1 limits gross chromosomal rearrangements (GCR) by inhibiting de novo telomere addition at double-stranded breaks[15,17]. GCR in yeast expressing Pif1 variants with a *hxt13:URA3* cassette 7.5 kb

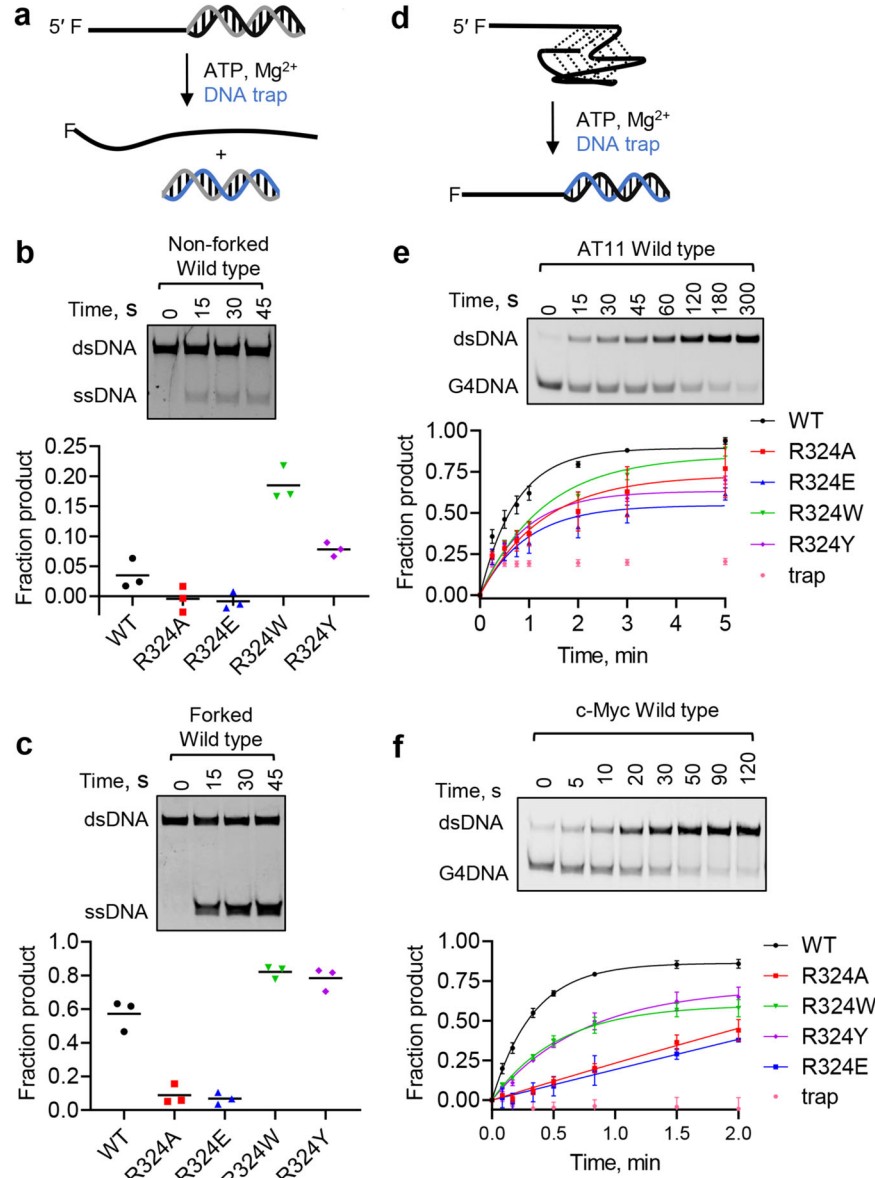

**Fig. 5 | R324 is involved in unwinding duplex DNA and unfolding G4 DNA.**
**a** Illustration of unwinding assay. Gels show unwinding under single-turnover conditions of a non-forked duplex (**b**) and forked duplex (**c**) by wild type ScPif1. Amplitudes of product formation for duplex DNA unwinding by ScPif1 variants is plotted. **d** Illustration of a trapping assay to measure G4 DNA unfolding. Gels show unfolding of a AT11 (**e**) and c-Myc (**f**) G4 DNAs under multi-turnover conditions by wild type ScPif1. G4 DNA unfolding by ScPif1 variants is plotted. Rate constants for unfolding are $1.4 \pm 0.3\,s^{-1}$, $0.81\,s^{-1}$, $1.1 \pm 0.4\,s^{-1}$, $0.78 \pm 0.12\,s^{-1}$, and $1.1 \pm 0.2\,s^{-1}$ for unfolding of AT11 by wild type, R324A, R324E, R324W, and R324Y Pif1, respectively. Rate constants for unfolding are $3.0 \pm 0.4\,s^{-1}$, $0.16\,s^{-1}$, $0.33 \pm 0.46\,s^{-1}$, $1.8 \pm 0.1\,s^{-1}$, and $1.4 \pm 0.2\,s^{-1}$ for unfolding of c-Myc by wild type, R324A, R324E, R324W, and R324Y Pif1, respectively. Results are average and standard deviation of triplicate independent experiments except G4 DNA unfolding of R324E, which is duplicate independent experiments. Source data are provided as a Source Data file.

telomeric to CAN1 on chromosome V was measured by simultaneous loss of the CAN1 and URA3 markers (Fig. 6d). GCR rates were high in cells lacking Pif1 (empty vector) or expressing ATPase deficient Pif1 (K264A). However, GCRs were suppressed in cells expressing wild type Pif1 and all of the R324 variants, suggesting that interactions of the residue at position 324 with the DNA are not necessary to suppress GCRs. Thus, a residue at position 324 that can interact with the DNA is necessary for Pif1 function in Okazaki fragment processing but not for mitochondrial respiration or suppression of gross chromosomal rearrangements. This suggests that robust helicase activity like that of wild type Pif1 is necessary for Okazaki fragment processing, but the limited helicase activity exhibited by R324A and R324E for both dsDNA and G4 DNA is sufficient for mitochondrial function and suppression of GCRs.

## Discussion

Pif1 is a potent G4 resolving helicase that preferentially unfolds antiparallel G4s over parallel G4s[22,34,46]. Several studies have investigated G4 DNA unfolding by ScPif1[34,44,46]. One report showed that ScPif1 uses its patrolling activity to unfold G4 structures in three discrete steps[47] while another report showed that ScPif1 unfolds G4 structures sequentially in two large steps[48], each of which melts one column of a quadruplex structure. Another showed that ScPif1 unfolds a G4 structure in 4–5 steps[44]. All reports suggest that G4 unfolding by a ScPif1 monomer involves a G3-triplex intermediate. Consistent with this finding, our ScPif1-G4 structure showed that a ScPif1 monomer recognizes and unfolds AT11 G4. This is dramatically different from dsDNA unwinding whereby Pif1 dimers are required for efficiently unwinding dsDNA[41,49,50].

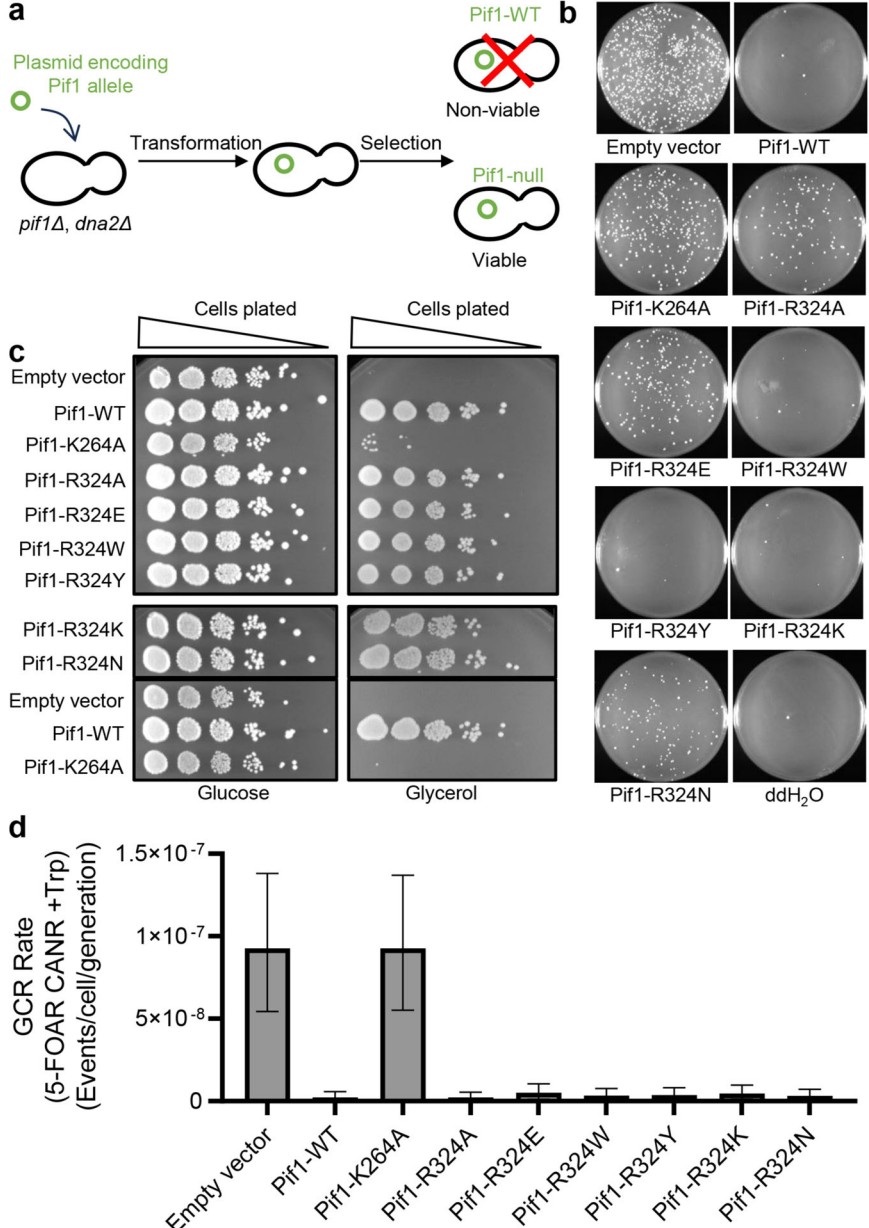

**Fig. 6 | Interactions of residue 324 with the DNA are critical for Pif1 function in Okazaki fragment processing but not for mitochondrial function or suppression of gross chromosomal rearrangements. a**, **b** Assay for Okazaki fragment processing. Adapted from Gao J, Proffit DR, Marecki JC, Protacio RU, Wahls WP, Byrd AK, Raney KD[63]. Translated and reproduced by permission of Oxford University Press. Translation Disclaimer: OUP is not responsible or in any way liable for the accuracy of the translation. The Licensee is solely responsible for the translation in this publication/reprint. **a** A plasmid encoding a wild type or mutant Pif1 allele was transformed into *pif1Δ dna2Δ* haploid cells. **b** Plates from the transformation of each Pif1 allele were imaged. Cells expressing wild type Pif1, R324W, R324Y, and

R324K are unable to suppress the lethality of *dna2Δ*, thus do not grow on selective plates. ATPase deficient Pif1 K264A, and wedge-altering R324A, R324E, and R324N grow on selective plates indicating null-Pif1 function. **c** To test for mitochondrial function, serial dilutions of cells were plated onto SD-Trp agar plates containing 2% glucose, and in parallel onto SG-Trp agar plates containing 3% glycerol. All tested Pif1-DNA interaction variants maintain mitochondrial respiration. **d** Rates of gross chromosomal rearrangements (GCR) (median and 95% confidence interval) are based on fluctuation analyses of frequencies of 16 individual cultures. Source data are provided as a Source Data file.

The structure of ToPif1-G4 showed that ToPif1 recognizes G4 predominantly through its 2B domain[31]. As the 2B domain is not very well conserved across Pif1 family members (Supplementary Fig. 4) and the wedge region has no contact with the bound G4, G4 recognition by ToPif1 appears to be unique to ToPif1 and cannot be broadly extended to other Pif1 family enzymes. Our ScPif1-G4 structure showed that the structurally conserved wedge region contacts the ssDNA/G4 junction with R324 playing a key role in mediating the interaction of ScPif1 with G4. Consistent with the structural observation, replacement of R324 by Ala or Glu reduced the G4 unfolding activities dramatically while

substitution of R324 with an aromatic residue (Trp or Tyr) decreased the G4 unfolding moderately. Our structural and mutational data suggest a mechanism governing AT11 G4 unfolding by ScPif1 (Fig. 7). One ScPif1 molecule binds to the AT11 G4 DNA (Fig. 7a), with the wedge region contacting the ssDNA/G4 junction (Fig. 7b). The binding of the G4 DNA causes a rotational conformational change of 2B domain to contact the ribose-phosphate backbone of the first G-tetrad to further enhance the interaction between ScPif1 and G4. R324 in the wedge region is anchored to G12 via cation-π interaction and engages the first guanine base G9 in the first G-tetrad indirectly through T8 in the ssDNA

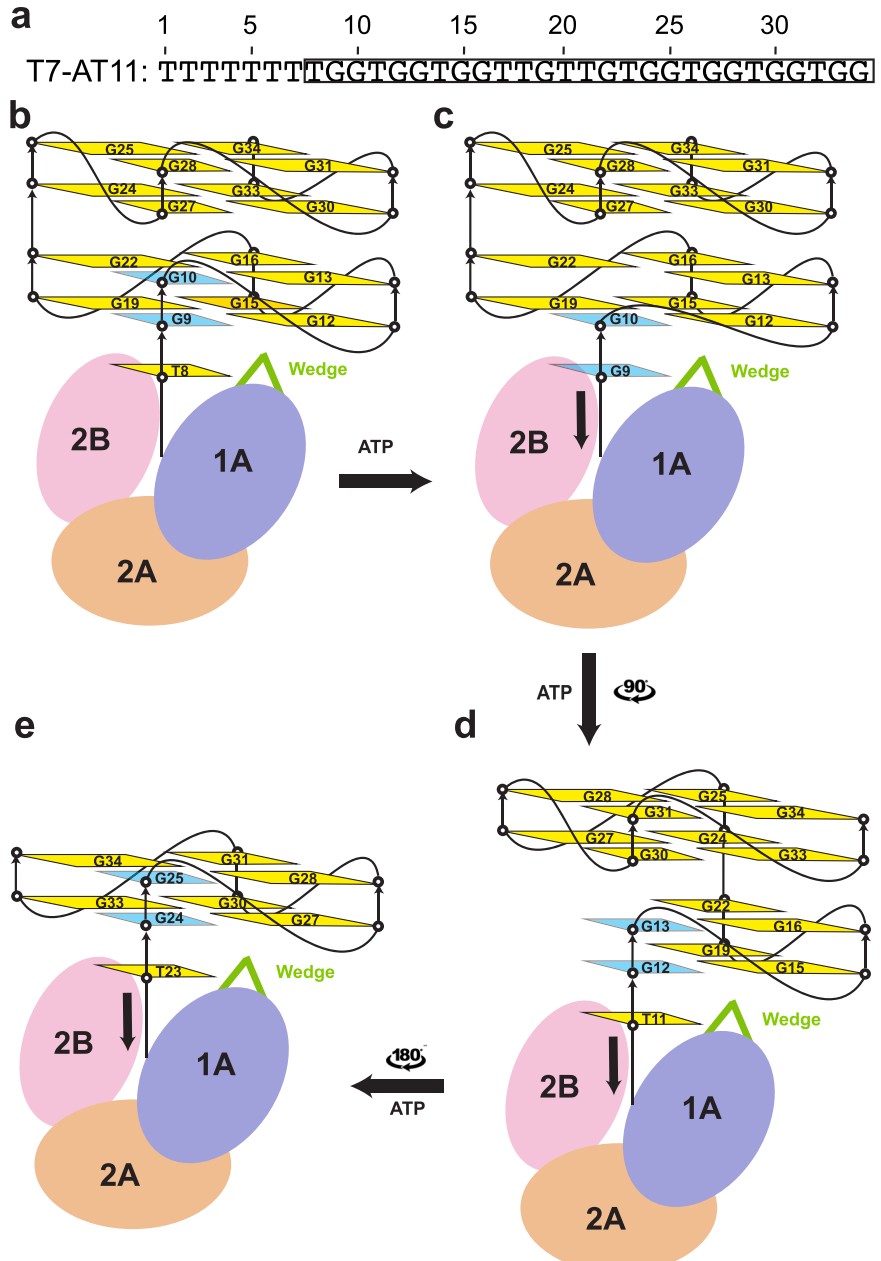

**a**
T7-AT11: TTTTTTT**TGGTGGTGGTTGTTGTGGTGGTGGTGG**

**Fig. 7 | Mechanism of G4 unfolding by ScPif1. a** The DNA sequence of T7-AT11. **b** AT11 is a four-layer G4 comprising of two parallel-stranded subunits connected by a central linker. One ScPif1 molecule binds to the T7-AT11 substrate DNA with the wedge contacting the ssDNA/G4 junction. **c** The interaction of the wedge with the first G-tetrad would deliver the first guanine nucleotide G9 into the ssDNA binding channel coupled with ATP hydrolysis-mediated translocation. **d** Further translocation of ScPif1 along ssDNA would move G10 to occupy the G9 position such that the G9-G10 column in the first and second G-tetrads would be stripped first to form the G3-triplex intermediate. **e** The remaining G-columns could be stripped one by one to complete the whole unfolding process.

region. The nature of R324 interacting with the G4 DNA suggests that it appears to be positioned to deliver the first guanine nucleotide G9 into the ssDNA binding channel coupled with ATP hydrolysis-mediated translocation (Fig. 7c). Further translocation of ScPif1 along ssDNA would move G10 to occupy the G9 position such that the G9–G10 column in the first and second G-tetrads would be stripped first to form the G3-triplex intermediate (Fig. 7d) as observed by single molecule FRET[47,48]. Then the remaining G-columns could be stripped one by one to complete the whole unfolding process (Fig. 7e). The G4 unfolding mechanism proposed here could be applied to unfolding of other parallel G4s (Supplementary Fig. 10a and b) as well as antiparallel G4s (Supplementary Fig. 10c and d) by Pif1.

Interestingly, the R324 variants unwind both forked and nonforked DNA duplexes with effects similar to those observed for the two G4 substrates AT11 and c-Myc (Fig. 5 and Supplementary Fig. 8). These observations suggest that ScPif1 unwinds dsDNA and unfolds G4 using similar mechanisms, in which the wedge region plays a key role. The actions of R324 and the Trp and Tyr variants are similar to the Phe residue of the pin region in another SF1B helicase Dda[39].

Several lines of evidence suggest that prokaryotic Pif1 helicases appear to use different mechanisms for dsDNA unwinding and G4 unfolding. First, unlike the wedge in ScPif1, the wedge in prokaryotic Pif1 helicases comprises an extended region followed by an α-helix[27] and no residue can be identified in prokaryotic Pif1 corresponding to

R324 except for K87 in BaPif1. Second, the structure of BaPif1 bound to a forked dsDNA showed that the wedge region is located far away from the ssDNA/dsDNA junction, therefore it is not involved in dsDNA unwinding. Third, the 5′-ssDNA tail bound to BaPif1 exhibited a sharp bent conformation while a ssDNA bound to ScPif1 has no such bent conformation[27,29]. The bend at the ssDNA/dsDNA junction has been proposed to contribute dsRNA unwinding for BaPif1[50] as proposed for PcrA, UvrD, and bacterial RecQ helicases[43,51,52].

Surprisingly, we found that interactions at residue 324 in ScPif1 are not necessary for respiratory proficiency or suppression of GCRs. However, these interactions at residue 324 are necessary for Pif1 function in Okazaki fragment processing. Previous reports of Pif1 variants that are functional in the mitochondria but not the nucleus either lacked a nuclear localization sequence[37] or had mutations in the nuclear localization sequence[53]. To our knowledge, these are the first ScPif1 variants that exhibit a defect only for Okazaki fragment processing. The reasons for this are not clear, but when ScPif1 creates a long flap at an Okazaki fragment, it likely requires unwinding of >25 base pairs. The R324A and R324E variants produced very little product from a forked duplex in a single-turnover (Fig. 5), suggesting that the interaction at residue 324 with the substrate is critical for processive unwinding of a duplex, which is similar to the activity of Pif1 at Okazaki fragments.

Our ScPif1-G4 structure and mutational data allowed us to propose a mechanism governing G4 unfolding wherein a G4 DNA would be unfolded in a sequential manner (Fig. 7), resulting in G3 triplex and G2 hairpin as intermediate states as observed by single-molecule FRET experiments[47,48]. This mechanism could explain how both parallel and antiparallel G4 structures are unfolded (Supplementary Fig. 10). Previous studies showed that ScPif1 preferentially unfolds antiparallel G4s over parallel G4s and its activity is modulated by thermal stability and loop lengths of G4 substrates[34,46]. The simple explanation for slow unfolding of a G4 DNA is that ScPif1 would pass through longer G-columns as the number of tetrads, and hence stability increases, and more ATP hydrolysis events would be required to destabilize more stable structures. In support of this notion, our G4 unfolding assays confirmed that ScPif1 indeed unfolds c-Myc G4 faster than AT11 (Fig. 5e, f).

DHX36 has been studied extensively for its G4 unfolding mechanism. A crystal structure of bovine DHX36 bound to a G4 DNA showed that upon DHX36 binding, the 5′ top G-tetrad has changed from a canonical to noncanonical G-quartet caused by the shifting of 3′ most guanine of the bottom G-quartet into the 3′ ssDNA region[13]. However, the resolution (3.8 Å) of the DHX36-G4 structure would not allow the unambiguous assignment of nucleotides in the G4 substrate. Furthermore, structural comparison between DHX36 with bound G4 and its DNA-free state showed that DHX36 undergoes conformational changes in the C-terminal and RecA2 domains. Structural information together with the smFRET data suggested that G4 DNA binding alone induces rearrangements of the helicase core, which results in the repetitive partial unfolding of G4 with one-base translocation. However, structural superposition of DHX36-G4 with DHX36 bound to a ssDNA showed that the conformations of DHX36 in the two structures are very similar (Supplementary Fig. 11), suggesting that the conformational changes of DHX36 upon G4 binding are induced by the 3′ ssDNA tail binding rather than the G4 binding. Moreover, Yan et al. showed that G4 is stabilized by DHX36 in its nucleotide-free state and is destabilized upon ATP hydrolysis[54]. Altogether, these results suggest that the mechanism by which DHX36 unfolds G4 still remains elusive, and further studies are required to clarify how ATP binding and hydrolysis are coupled with G4 unfolding.

In conclusion, our studies have identified a mechanism by which ScPif1 uses a conserved wedge region to unwind dsDNA and to unfold G4 DNA. This mechanism is likely shared by other eukaryotic Pif1 helicases but might be different from that used by prokaryotic Pif1 helicases. The mechanism is strikingly different from those proposed for DHX36 and RecQ[55] that harbor a G4-specific recognition motif and guanine-specific binding pocket, respectively. As many helicases both unwind dsDNA and unfold G4 DNA, it remains to be seen whether the G4 unfolding mechanism we propose here can be generalized to other helicase families with both duplex and G4 DNA unwinding activities.

## Methods

### Identification of AT11 as the G4 DNA substrate for ScPif1
The DNA samples were purchased from Integrated DNA Technologies and dissolved in a buffer solution containing 70 mM KCl, 20 mM potassium phosphate (pH 7.0), 10% $D_2O$, and 20 μM DSS. DNA concentrations of 100 μM and 5 μM were used for all the NMR and CD experiments respectively. The sequences were annealed by heating the DNA samples at 95 °C for 5 min in a water bath and letting them cool down slowly to room temperature. All the NMR experiments were performed on 600 MHz Bruker AVANCE II spectrometer at 25 °C using the modified jump-and-return pulse program. DNA concentration used was 100 μM. All spectra were processed and analyzed using TopSpin 4.0.6. The circular dichroism (CD) spectra of all the DNA samples were measured with JASCO-815 spectrophotometer. The respective DNA samples with bound protein (used for NMR) were diluted to a final concentration of ~4–5 μM in the buffer solution. Subsequently, 500 μl of the diluted DNA sample was transferred to a 1 cm path-length cuvette and the CD spectrum was recorded at 25 °C with 10 accumulations. The data was baseline corrected from the signal contributed by the buffer. Each spectrum was further corrected to be zero at 320 nm wavelength.

### The G4 DNA substrate T7-AT11 preparation for crystallization
The G4 DNA substrate T7-AT11 (TTTTTTTTGGTGGTGGTTGTTGTGG TGGTGGTGGT) was purchased from Integrated DNA Technologies and dissolved in a buffer containing 70 mM KCl and 20 mM potassium phosphate (pH 7.0) to achieve a final concentration of 0.1 mM. The solution was then heated to 95 °C and gradually cooled to room temperature in a water bath to induce the annealing of the substrate. Subsequently, T7-AT11 was further purified by gel filtration on a Superdex 75 column (GE Healthcare) equilibrated with 20 mM Tris-HCl, pH 7.5, 150 mM KCl, 5% glycerol, 1 mM DTT.

### Protein purification and crystallization
To produce the *Saccharomyces cerevisiae* Pif1 (Uniprot: P07271) protein and its mutants (R324A, R324E, R324W and R324Y) for crystallization and in vitro assays, a truncated ScPif1 (residues 236–753) containing the helicase core domain was cloned into a pET28a vector with an N-terminal SUMO tag and expressed in *E. coli* BL21 cells.

Point mutants of ScPif1 were created in the pET28a vector using site-directed mutagenesis method.

Cells were harvested (5000 × g, 20 min, 4 °C), resuspended in buffer A (20 mM Tris-HCl, pH 7.5, 1 M NaCl, 5% glycerol, 1 mM DTT), and lysed by sonication. The lysate was cleared by centrifugation (18000 × g, 1 h, 4 °C), and the supernatant was incubated with Ni-NTA resin (GE Healthcare). Bound protein was washed with buffer A, eluted with buffer B (20 mM Tris-HCl, pH 7.5, 1 M NaCl, 5% glycerol, 1 mM DTT, 300 mM Imidazole). The SUMO tag was cleaved from the protein by Ulp protease and the protein was buffer-exchanged to buffer C (20 mM MES, pH 6.5, 150 mM NaCl, 5% glycerol, 1 mM DTT) and applied to Mono S Column (GE Healthcare), eluted with a gradient from 150 mM to 1 M NaCl and further purified by gel filtration on a Superdex 200 column (GE Healthcare) in buffer D (20 mM Tris-HCl, pH 7.5, 500 mM KCl, 5% glycerol, 1 mM DTT). The ScPif1 mutants were purified using the same protocol as that used for wild type ScPif1.

The purified ScPif1 was incubated with the T7-AT11 at a molar ratio of 1:1.2 in the presence of ATP transition-state analog

**Table 1 | Data collection and structure refinement statistics**

| Data collection | |
|---|---|
| Resolution (Å) | 48–3.5 (3.625–3.5) |
| Space group | C2 |
| Cell dimensions | |
| a, b, c (Å) | 56.163, 125.141, 258.233 |
| α, β, γ (°) | 90.000, 91.467, 90.000 |
| $R_{merge}$ (%) | 0.2202 (0.5099) |
| I/σI | 3.05 (1.49) |
| CC1/2 | 0.904 (0.611) |
| Completeness (%) | 99.35 (97.57) |
| Redundancy | 1.9 (1.8) |
| Refinement | |
| No. reflections | 22444 (2168) |
| $R_{work}$/ $R_{free}$ | 0.2815 (0.3292)/0.3199 (0.3829) |
| No. atoms | |
| Protein and DNA | 9300 |
| Ligand/ion | 72 |
| B-factor (Å$^2$) | |
| Protein and DNA | 47.44 |
| Ligand/ion | 39.15 |
| R.M.S. deviations | |
| Bond angles (°) | 1.18 |
| Bond length (Å) | 0.005 |

Values in parentheses are for highest-resolution shell.

ADP·AlF$_4^-$. The resulting ScPif1-G4 complex was purified by gel filtration on a Superdex 200 column (GE Healthcare) in buffer E (20 mM Tris-HCl, pH 7.5, 150 mM KCl, 5% glycerol, 1 mM DTT) and then concentrated to ~10 mg·ml$^{-1}$. The purified ScPif1-G4 complex was subjected to crystallization screening at 20 °C using the sitting drop vapor diffusion method. Crystals of ScPif1-G4 were obtained in a condition consisting of 0.1 M MES pH 6.9, 22.5%(w/v) PEG 400, and 0.1 M sodium acetate.

### CD spectroscopy of the wild type ScPif1 and its variants
CD spectra were measured on a Jasco J-810 spectrophotometer at a resolution of 0.1 nm, with a bandwidth 2 nm. Wavelength scans were averages of five scans between 200 nm and 300 nm collected at 20 °C in a quartz cuvette with a 1-mm path length with 5 μM proteins. Mean residue ellipticity was calculated according to the CD measurement manual.

### Structure determination of ScPif1-G4
X-ray diffraction data were collected at the Australian Synchrotron beamline (MX2). The crystal structure of ScPif1-G4 complex was determined by successive rounds of molecular replacement with PHASER 3 using ScPif1 (PDB code: 5O6D) and AT11 (PDB code: 2N3M) as search models[56]. The final model was built to acceptable stereochemical values with iterative model building and refinement cycles using CCP4 Program Suite v8.0.019[57] and Phenix version 1.21[58]. Data collection and refinement statistics are given in Table 1.

### Oligonucleotides
Oligonucleotides (Supplementary Table 1) for kinetic assays were purchased from Integrated DNA Technologies and resuspended in 10 mM Tris pH 7.5, 1 mM EDTA. Labeled strands (1 μM) were mixed with 1.2 μM unlabeled complementary strands (if applicable) in 10 mM Tris pH 7.5, 1 mM EDTA, 100 mM KCl. Samples were heated to 95 °C for 10 min and slowly cooled to room temperature to allow folding of duplex and G4 structures.

### Fluorescence anisotropy binding assay
The measurements were carried out using a plate reader (Tecan) at 20 °C. FAM-labeled AT11 was added to each reaction at a fixed concentration of 10 nM with different concentrations of wild type ScPif1 or its mutants in 100 μl buffer containing 20 mM HEPES-NaOH pH 7.5, 100 mM NaCl, 10% Glycerol. Prior to measurement, the reaction mixtures were equilibrated for 30 min. Triplicate measurements were performed for each titration point, with 485 nm and 535 nm as the excitation and emission wavelength, respectively. The data were analyzed using the quadratic binding equation[59] with Prism 9 (GraphPad).

### Duplex DNA unwinding
All concentrations listed are final, after mixing. Substrate (20 nM) and Pif1 (200 nM) were pre-incubated at 25 °C for 5 min in assay buffer (25 mM HEPES pH 7.5, 50 mM KCl, 2 mM β-mercaptoethanol, 0.1 mM EDTA, 0.1 mg/mL BSA). In another tube, 5 mM ATP, 10 mM MgCl$_2$, 600 nM duplex trap complementary to the displaced strand, and 20 μM T$_{50}$ were mixed in assay buffer and pre-incubated at 25 °C for 5 min. Single-turnover unwinding reactions were initiated by mixing the enzyme/DNA solution with the ATP/MgCl$_2$ solution. T$_{50}$ was added with the ATP as a protein trap to prevent Pif1 from rebinding to the substrate after dissociation. Samples were removed and quenched with 200 mM EDTA, 0.6% sodium dodecyl sulfate, 0.1% Orange G, and 6% glycerol. Samples were separated by 20% native PAGE and visualized using an Amersham Typhoon RGB imager and quantified to determine the fraction ssDNA product.

### Trapping assay for G4 DNA unfolding
All concentrations listed are final, after mixing. Substrate (20 nM) and Pif1 (200 nM) were mixed in assay buffer (25 mM HEPES pH 7.5, 50 mM KCl, 2 mM β-mercaptoethanol, 0.1 mM EDTA, 0.1 mg/mL BSA) immediately before addition of a solution containing 5 mM ATP, 10 mM MgCl$_2$, and 600 nM trap complementary to the G4 forming region. Samples were removed and quenched with 200 mM EDTA, 0.6% sodium dodecyl sulfate, 0.1% Orange G, 6% glycerol, 4 μM T$_{50}$, and 20 μM C trap complementary to the trapping strand to prevent the trapping strand from spontaneously unfolding the G4 structure after quenching the reaction. Samples were separated by 20% native PAGE, visualized using an Amersham Typhoon RGB imager, and the fraction ssDNA product at each time point was determined. Data were fit to a single exponential function using Prism 9 (GraphPad).

### Reporter assay for G4 DNA unfolding
All concentrations listed are final, after mixing. Substrate was prepared with a 5′-ssDNA overhang c-Myc G4 followed by a 3′-tailed 12 base pair duplex to serve as a reporter for G4 DNA unfolding (Supplementary Table 1 and Supplementary Fig. 8a)[34]. Substrate (20 nM) and Pif1 (200 nM) were mixed in assay buffer (25 mM HEPES pH 7.5, 50 mM KCl, 2 mM β-mercaptoethanol, 0.1 mM EDTA, 0.1 mg/mL BSA) immediately before addition of a solution containing 5 mM ATP, 10 mM MgCl$_2$, and 600 nM duplex trap complementary to the displaced strand. Samples were removed and quenched with 200 mM EDTA, 0.6% sodium dodecyl sulfate, 0.1% Orange G, 6% glycerol, and 4 μM T$_{50}$. Samples were separated by 20% native PAGE, visualized using an Amersham Typhoon RGB imager, and the fraction ssDNA product at each time point was determined. Data were fit to a single exponential function using Prism 9 (GraphPad).

### Yeast strains and plasmids
*S. cerevisiae* strains for measuring Pif1 function in vivo were kind gifts from Dr. Virginia Zakian: MBY77 (YPH500, hxt13:URA3, pif1::HIS3MX6) and YCG59 (W303, PIF1/pif1::NatMX6, DNA2/dna2::KanMX6)[37]. The haploid strain G0339 (*W303, pif1::NatMX6, dna2::KanMX6*) was generated by tetrad dissection of YCG59 asci. The yeast strains are described in Supplementary Table 4.

Plasmids for expression of full-length ScPif1 in *S. cerevisiae* under control of the *PIF1* promoter were gifts from Dr. Virginia Zakian: pMB13 (pRS414 empty vector), pCG17 (pRS414-Pif1wt-3xFLAG), and pCG18 (pCG17-Pif1-K264A)[37]. Site-directed mutagenesis was used to create plasmids encoding each of the Pif1 variants (Supplementary Tables 3).

### Okazaki fragment processing

Pif1 function in Okazaki fragment processing was measured as suppression of *dna2Δ* lethality as previously described[37]. Each pCG17-serial plasmid encoding Pif1 (WT or mutant) or a pMB13 empty vector control (1 μg) was transformed into haploid strain G0339 (*W303, pif1::NatMX6, dna2::KanMX6*) by the lithium-acetate-method[60]. Approximately $1.5 \times 10^8$ early-log-phase cells were transformed without carrier-DNA and plated onto SD-Trp medium. After incubation at 28 °C for 5 days, plates were imaged. Because loss of Pif1 function suppresses the lethality of *dna2Δ*, growth of colonies indicated that the Pif1 mutant was a null allele in this assay.

### Mitochondrial function

Mitochondrial function was assessed by monitoring the ability of colonies to grow on plates containing glycerol. Each pCG17-serial plasmid encoding Pif1 (WT or mutant) or an empty vector control (pMB13) was transformed into a heterozygous diploid strain YCG59 (*PIF1/pif1::NatMX6, DNA2/dna2::KanMX6*)[37] by the lithium-acetate-method. After sporulation and tetrad dissection, spore colonies were genotyped. Haploid *DNA2+ pif1::NatMX6* spores carrying plasmids were selected by their *KanMXS NatMXR Trp+* phenotype. These selected haploid cells were grown at 28 °C overnight in SD-Trp (10 mL) to mid-log phase. Approximately $1.2 \times 10^7$ cells were diluted to 200 μL using water in a 96-well microplate and diluted with 10-fold dilutions from 100 to $10^{-5}$ using sterile water. Approximately 3 μL of each dilution was spotted onto SD-Trp agar plates containing 2% glucose or 3% glycerol. Glucose plates were grown at 28 °C for 3 days, and glycerol plates were grown for 5 days.

### Gross chromosomal rearrangement (GCR)

Experiments were performed as described previously[37]. Briefly, two independent isolates of each Pif1 allele were separately streaked onto SD-Trp agar plates and grown at 28 °C for 4 days. Eight clones of each isolate were separately inoculated in 5 ml of SD-Trp and grown at 28 °C for 3 days. Dilutions of each culture (50 μl of a 1:10,000 dilution) were plated on non-selective SD-Trp plates and incubated at 28 °C for 3 days. GCRs were identified on SD-Trp with 1 mg/mL 5-fluoroorotic acid and 50 μg/mL Canavanine Sulfate (SD-Trp + FOA + CAN) plates. Selective plates had 1 ml of each culture pelleted and resuspended in 100 μl sterile $H_2O$ before plating and incubating at 28 °C for 7 days. GCR rates were determined using the method of median and 95% confidence intervals were calculated by FluCalc version 2019.4.1[61]. At least 16 individual cultures were used for each rate determination.

### Western blotting

Total protein was extracted from *S. cerevisiae* using a procedure previously optimized for yeast[62]. Briefly, yeast strains carrying pCG17-serial plasmids that encode WT or mutant PIF1 alleles or pMB13 (empty vector control) were grown in SD-Trp + Ade (10 ml) at 28 °C overnight. After pelleting, cells were resuspended in 600 μL of alkaline lysis buffer (0.1 M NaOH, 50 mM EDTA, 2% SDS, 2% β-mercaptoethanol) and incubated for 10 min at 90 °C. Samples were neutralized by addition of 15 μL of 4 M acetic acid to each sample, mixed by vortexing for 30 s, and incubated for 10 min at 90 °C. The concentration of protein in each lysate was measured using a Coomassie Plus Protein Assay Kit (Pierce). Proteins (18 μg) were separated by 8% SDS-PAGE and transferred to a 0.45 μm nitrocellulose membrane at 4 °C. The membrane was initially stained with 0.11% Ponceau S in 5% acetic acid, and total protein was imaged on a Bio-Rad ChemiDoc MP Imaging system. The

Ponceau was removed by washing twice in TBS-T before blocking with 3% BSA in TBS-T for 2 h at room temperature. The membrane was incubated with mouse anti-FLAG M2 (Sigma F1804; 1:1000) in TBS-T with 3% BSA overnight at 4 °C. After washing, the membrane was incubated with HRP-labeled goat anti-mouse IgG (PerkinElmer EF822001EA, 1:10,000) in TBS-T with 3% BSA for 2 h at room temperature. After washing, the membrane was incubated with Amersham ECL Prime Western Blotting Detection Reagents (Cytiva RPN2232). The membrane was imaged using a Bio-Rad ChemiDoc MP Imaging system.

### Reporting summary

Further information on research design is available in the Nature Portfolio Reporting Summary linked to this article.

## Data availability

The atomic coordinates and structural factors for the ScPif1-G4 complex have been deposited with the Protein Data Bank under accession code 8XAK. The atomic models used in this study are available in the Protein Data Bank under accession codes 7OAR, 5VHE, 5FHH, 5FHE, 5O6B, 3GPL, 5N8S, 5O6D, 2N3M. Source data are provided with this paper.

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

## Acknowledgements

We would like to thank the beamline scientists at MX2 of the Australian Synchrotron Radiation Facility for assistance of data collection. This work was supported by grants from the US National Institutes of Health (P20GM121293; R35GM122601 to K.D.R; and GM145834 to W.P.W. and the Agency for Science, Technology and Research in Singapore.

## Author contributions

H.S., A.K.B., and K.D.R. conceived and supervised the project. Z.H. and V.Q.T. performed protein purification, crystallization, structure determination, and DNA binding experiments. P.D. performed NMR and CD spectroscopic experiments. A.K.B., E.G.M., B.O., J.G., and J.C.M. performed the in vitro biochemistry experiments. J.G. and R.U.P. constructed yeast strains. J.G. performed the in vivo experiments. W.P.W. analyzed the in vitro biochemical data. H.S., A.K.B., and K.D.R. wrote the manuscript. All authors contributed to data interpretation and editing the manuscript.

## Competing interests

The authors declare no competing interests.
