## [Peer Review File · Nature Communications]

Eukaryotic Pif1 helicase unwinds G-quadruplex and dsDNA using a conserved wedgeREVIEWER COMMENTS

Reviewer #1 (Remarks to the Author):

In the submitted manuscript Hong et al present a new crystal structure of a yeast Pif1 DNA helicase bound to a DNA substrate with a 5'-ssDNA tail and a downstream G-quadruplex, that is resolved in the structure. The authors also present mutational studies of a conserved Arg, that in the structure makes direct contact with the G-quadruplex, and show that while in vitro mutations at this position affect both G-quadruplex and dsDNA unwinding, in vivo the same mutation only affect Okazaki fragment processing but not mitochondrial functions or gross chromosomal rearrangement. Overall, this work complements and expands previous structural studies on the Pif1-family of helicases and provides a novel structure of Pif1 bound to a G-quadruplex, distinct from a previous one obtained using *Thermus Oshimai* Pif1.

1) The authors present binding studies in Supplementary Figure 1, and in the main text they refer to these as reporting on Pif1 binding to the G-quadruplex. However, inspection of the DNA sequences in Supplementary Table 1 shows that all the DNA substrates used for these studies contain a 5' oligo dT ssDNA tract. As such, the binding data represent a more complex binding system than the reader is led to believe by the authors' description. Using the substrates indicate in Supplementary Table 1, the binding is a convolution of the interaction of Pif1 with the 5' ssDNA region and the G-quadruplex. Please, address this point by clearly stating that the binding data represent more than just binding to the G-quadruplex, or preferably include experiments using ssDNA and a plain G-quadruplex (no ssDNA extension) to address this issue.

2) Please clearly state in the Results section that the structure was obtained with a truncated version of Pif1. This is explicitly state for the DNA binding studies and in Methods, but it should also made clear in the Results/Discussion.

3) From the in vitro data mutations R324A and R324E affect both dsDNA and G-quadruplex unwinding, yet in vivo these mutations seem to specifically affect Okazaki fragment maturation (or more simply, suppression of *dna2Δ* lethality) while they do not seem to have a role in Pif1 mitochondrial function or GCR. Do the author suggest that dsDNA/G-quadruplex are not required for the latter? As stated by the authors, these findings are surprising, and they deserve more explanation/interpretation than what provided in the Results or Discussion.

4) While the authors present both DNA binding and unwinding data for the R324 mutations, no data testing whether possible differences in ATPase activity can explain the behavior of R324A/E mutants. Please include ATPase data for the mutants used.

Reviewer #2 (Remarks to the Author):

Summary:

In this manuscript, Hong et al. reported their structure-function analysis work on the X-Ray crystal structure of the yeast (*S.cerevisiae*) Pif1 helicase bound to a DNA G-quadruplex (G4). The 3.5 Å structure provided insights into how scPif1 interacts with a DNA G4 molecule and the authors' interpretation of the protein-nucleic acid and protein-protein interactions are tested in biochemical and yeast cellular assays. The authors compared their structure with existing similar structures of Pif1 from other species and deduced that the scPif1 engages G4 using a different mechanism via its wedge region, instead of the 1/2B domains utilized by the ToPif1. The authors focused on an arginine residue (R324) that they identified from their structure and tested its variants' impact on dsDNA and G4 unwinding using in vitro biochemical assays. They found R324 variations had an impact on the helicase unwinding of dsDNA and G4. Further testing of the variants' impact on Pif1's functions in yeast cells yielded a range of effects; R324 is important for Okazaki fragment processing but not for Pif1's roles in respiratory proficiency or suppression of chromosomal rearrangements.

While the X-ray crystal structure of scPif1-G4 and the ensuing biochemical/cellular data are highly informative and interesting, I have several concerns that I believe the authors should address before

this body of work is ready for publication. For details, please see the following comments:

Comments:

1. Page 5, line 93-94: The authors stated that the CD spectrum of scPif-G4 and the G4 are similar. But Fig. S1b is not showing the two spectra are the same. It will be extremely helpful and less confusing to mention how the authors interpret the CD spectra and deduce that the AT11 G4 is still intact (i.e, a negative 240 nM band and a positive 260 nM peak are characteristics of a parallel G4 CD spectrum).
2. Page 5, line 112: I highly recommend adding a zoom-in inset for Fig S3 to help the readers see the 2B domain differences between the two structures. In the current version, the figure is cluttered and difficult for readers to find the 2B domain annotation.
3. Page 5, line 116: Figure 1b will benefit from adding more figure panels illustrating different views of the said interactions and include zoom-in insets of various key interactions, especially that of R324. The authors can provide dashed yellow lines connecting two interacting residues to provide the premises to their interaction analysis.
4. Page 6, line 129-131: the FP binding assay setup is in the titration regime; the probe concentration (10 nM) is similar to the measured Kd values (15-40 nM). Fitting the data with the Hill equation is not adequate in this condition. The quadratic binding equation will be more appropriate. Furthermore, it is unclear if the reactions have reached binding equilibrium (incubation time is not mentioned). I highly encourage the authors to follow the recommendations from Jarmoskaite et al., 2020 eLife to establish the appropriate experimental setup and analysis to measure binding affinities. The R324 variants' binding affinities play an important part in shaping the authors' conclusions.
5. Page 6, line 136-138: "These observations strikingly differ from those reported..". A figure showing this structural difference mentioned by the authors will be helpful for readers' understanding. And why is this difference important?
6. Page 6-7, line 144-145: "The stabilized wedge by the Pif1 signature motif would maintain its rigidity for exerting its helicase activity." It is unclear how a rigid wedge can support the Pif1 helicase activity. Please elaborate on this point, preferably with a supplementary figure. This will be extremely helpful for readers not in this area of research.
7. Page 7, line 154-155: "The striking difference between ScPif-G4 and ToPif1-G4 is the G4 orientation". Perhaps, the authors can elaborate on if the differences in the G4 DNA sequences used in the two structures would impact the way the two helicases recognize a G4 structure.
8. Page 7-8, line 167-177: This section/paragraph is confusing. The section title is the wedge region is a conserved feature. The focus is on R324 of scPif1. The authors described similar residue was found in several species' Pif1 sequences but not for some. So, do they think the wedge is still conserved? If R324 is not conserved in ToPif1, would the earlier comparison of scPif1 and ToPif1 suggest they ToPif1 uses a different mechanism to engage G4?
9. Page 8, line 181: Fig S6 is not helpful in seeing what the authors said in the main text. I recommend the authors provide a few more panels and maybe change the colors to better help the readers compare the two aligned structures.
10. Page 8, line 189-191: "...ScPif1 variants that retain the ability to interact with the G-tetrad (R324W and R324Y)...". How do the authors know that these variants still engage the G-tetrad as shown in the WT structure? If they have further structural data supporting these claims, please provide them. If not, the biochemical assays are only saying these variants still bind to the DNA molecule.
11. Page 8, line 191-193: "...R324W and R324Y ScPif1, which would be predicted to stack with G12 in the 5' most G-tetrad..". It will be extremely helpful for the readers if the authors elaborate why the tryptophan and tyrosine variants will stack with the G12. Are there reported precedents in similar structures? If yes, please include them in the main text.
12. Page 9, line 208-209: Please provide quantification of these rates.
13. The cellular assay results are interesting. Perhaps, it will be helpful if the authors can elaborate why they performed these assays, which will be helpful for the readers to understand the motivation tied to the R324 premise. For example, is the DNA-binding or helicase activity of Pif1 critical for these functions? If yes, how would the authors explain the non-effective impact of the tested variants on mitochondria respiratory function and GCR?

14. Page 11, section "ScPif1 unfolds G4 and unwinds dsDNA via a similar mechanism": This section should be under the discussion section.

Minor comments:

1. Page 3, line 56: Please provide the reference(s) for the said smFRET analysis. It is unclear if this information belongs to the reference from the prior sentence.
2. Page 4, line 70-73: I am confused to why the ToPif1 2B domain-G4 interaction provided little information. Maybe the ToPif1 uses a different mechanism from the wedge region that was proposed to be used by other species? There is no introduction to the wedge region before this sentence. Readers will benefit from a sentence or two introducing the wedge region (with references).
3. Page 6, line 124: Figure 2a is not very helpful for the readers to follow the description in the main text. I recommend adding a few more zoom-in insets detailing the interactions that the authors described in the main text.
4. Page 6, line 144: Fig S5 will benefit from adding interaction details; show contacts by dashed yellow lines between interacting residues.
5. Page 7, line 146-148: A 15 aa swap to alanine is an arguably disruptive mutagenesis approach. It is unclear if the wedge region has become more flexible due to the mutations, or the mutations have perturbed more beyond the flexibility of the wedge. Can the authors elaborate more on how the cited work supports their interpretation?
6. Page 7, line 164-164: Please add the references to either the manuscript figures or published work.
7. Page 10, line 227-230: This sentence is too long and confusing. Maybe break into two-three sentences?
8. Page 12, line 279-80: The authors may be intending to use a comma instead of a full stop after "(Fig. 7a)".
9. Page 16, line 370: "In conclusion, our studies have identified a unified mechanism by which ScPif1 uses a 371 conserved wedge region to unwind dsDNA and to unfold G4 DNA." I think this a unified mechanism is somewhat overstating given this statement is supported by a single structure from a single species.
10. No visible error bars for certain datasets in Figure 5e, f. For example, the trap data points do not seem to have error bars plotted in Figure 5e.
11. Page 4, line 92, please define "antiproliferative DNA sequence".
12. Most published PDBs do not have their accession numbers cited in their corresponding figure legends or text.
13. Describe how the authors perform sequence alignment to arrive to their conclusions.
14. Figure 5b graph is not making use of the space above. I think it will be easier for readers to appreciate the differences (and P values) if they constrain the Y-axis range to between [0, 0.25].

Reviewer #3 (Remarks to the Author):

The Article from Hong and colleagues with the title " Eukaryotic Pf1 helicase unwinds G-quadruplex and dsDNA using a unified mechanism" is written well however lacks details which makes it hard to follow the conclusions.

In this article they show a co-crystal of *S. cerevisiae* Pif1 DNA helicase with a bound G4 structure. They defined in biochemical experiments the interaction domains of Pif1 with the G4 itself, via the wedge motif. As Pif1 is a multifunctional helicase they characterized further this interaction for Pif1 function in living cells. Further they defined which AS is essential for Okazaki fragment processing. This AS is not important for Pif1 function in the mitochondria or gross chromosomal rearrangements.

Major comments

- Why not a natural G4 from *S. cerevisiae* was used as this would far be more realistic and

informative, also more data on how they identified AT11 (screen? Nature) and what AT11 is, is required (sequence, composition)?

- Figure 1a, why truncated ScPif1 and not full length? What is the truncation and why is this needed? The truncated version is not introduced. Same A1F4- is not introduced for Figure 1. Is Pif1 truncated in this crystal?

- They used in Figure 1 a 5' tailed G4, how long is the tail and why not a G4 without a tail as published previously was used.

- Also the complex description is very short and lacks details. Is Pif1 binding as a homodimer? As they state in line 99 : ' the symmetric unit of the crystal structure contains two essentially identical ScPif1 G4 complexes...' What does a rotation of 2B by 90 means?

- They nicely defined the R324 and R326, and K686 and p687 as interacting domains to the G4 of AT11. As Pif1 has different unwinding ability for parallel and anti-parallel is the binding different? How specific is the interaction for this AT11? Here more details on different binding of Pif1 to parallel vs antiparallel is needed. Further which G4 was used in the ToPif1 study, which topology had this G4? Is therefore the interaction different (see line 160 p7).

- P6 lines 162 following, information on DHX36 are missing, what is the DSM. Is this helicase truncated? Which G4 structure, topology, conformation? As DHX36 is a 3'-5' and Pif1 a 5'-3' how easy can this be compared? As DHX36 unfold DNA and RNA helicase it would be better to compare BLM or WRN at G4s. Therefore, the conclusion, that Pif1 unfolds differently is OK, but this is expected based on the function and target choice of DHX36 vs Pif1. The strong statement of the authors is too strong and is not holding. Is this true also for other G4 structures? Is Pif1 unique in comparison to other G4 unfolding helicases as it is so far the most efficient G4 unfolding helicase? Here more work is essential.

- Further details are missing on the G4 used in the reporter assay? Does this differ with different G4 substrates? What is with RNA/DNA hybrids?

- Further also the in vivo experiments are not written detailed enough to understand the experimental setup. Under which promoter is Pif1 expressed in this assay? Where are the Western confirming the expression? Same is true for the mitochondria experiments, is Pif1-from the plasmid expressed in the mitochondria? Or is the truncated version not entering the mitochondria? Same, the GCR rates are done well, however what kind of GCR event was obtained, here Southern blots are needed. And why where no experiments done to test G4 function in cells with the Pif1 variants? (Pif1 DNA binding? GCR at G4s or G4 levels in the cell?)

-

REVIEWER COMMENTS

Reviewer #1 (Remarks to the Author):

In the submitted manuscript Hong et al present a new crystal structure of a yeast Pif1 DNA helicase bound to a DNA substrate with a 5'-ssDNA tail and a downstream G-quadruplex, that is resolved in the structure. The authors also present mutational studies of a conserved Arg, that in the structure makes direct contact with the G-quadruplex, and show that while in vitro mutations at this position affect both G-quadruplex and dsDNA unwinding, in vivo the same mutation only affect Okazaki fragment processing but not mitochondrial functions or gross chromosomal rearrangement. Overall, this work complements and expands previous structural studies on the Pif1-family of helicases and provides a novel structure of Pif1 bound to a G-quadruplex, distinct from a previous one obtained using *Thermus Oshimai* Pif1.

Thanks for your helpful suggestions.

1) The authors present binding studies in Supplementary Figure 1, and in the main text they refer to these as reporting on Pif1 binding to the G-quadruplex. However, inspection of the DNA sequences in Supplementary Table 1 shows that all the DNA substrates used for these studies contain a 5' oligo dT ssDNA tract. As such, the binding data represent a more complex binding system than the reader is led to believe by the authors' description. Using the substrates indicate in Supplementary Table 1, the binding is a convolution of the interaction of Pif1 with the 5' ssDNA region and the G-quadruplex. Please, address this point by clearly stating that the binding data represent more than just binding to the G-quadruplex, or preferably include experiments using ssDNA and a plain G-quadruplex (no ssDNA extension) to address this issue.

We agree that it is important to clarify what substrates are used. We have listed all the DNA sequences used in crystallization, binding assay, unwinding and unfolding experiments in Supplementary Table 1. We have also added an explanation of why we used a 5'-tailed G4DNA to p5, lines 95-99 that is summarized here. Within a cell, G4DNA structures that are not located at the ends of telomeres will have overhangs on both the 5'- and 3'-end because G4s are more likely to form in ssDNA than dsDNA and because G4 formation destabilizes nearby duplexes. Because helicases have a channel for binding ssDNA, studying interactions of helicases with G4DNA with a ssDNA overhang on the appropriate side based on the direction of translocation will likely provide the most biologically relevant information. In addition, we have included the binding data of wild type ScPif1 to T10-FAM (ssDNA) and AT11-FAM (plain G-quadruplex) in Supplementary Figure 1c and reperformed the binding assays of AT11-FAM to wild type ScPif1 and its variants as suggested by the Reviewer 2 (Supplementary Figure 1e).

2) Please clearly state in the Results section that the structure was obtained with a truncated version of Pif1. This is explicitly state for the DNA binding studies and in Methods, but it should also made clear in the Results/Discussion.

Thank you for this suggestion. We have added to page 5, line 105 that the structure was obtained with the helicase domain of ScPif1.

3) From the in vitro data mutations R324A and R324E affect both dsDNA and G-quadruplex unwinding, yet in vivo these mutations seem to specifically affect Okazaki fragment maturation (or more simply, suppression of dna2Δ lethality) while they do not seem to have a role in Pif1 mitochondrial function or GCR. Do the author suggest that dsDNA/G-quadruplex are not required for the latter? As stated by the authors, these findings are surprising, and they deserve more explanation/interpretation than what provided in the Results or Discussion.

This is an important point. No, we do not suggest that neither dsDNA unwinding nor G4DNA unfolding are required for mitochondrial function or GCR suppression. Although R324A and R324E Pif1 have reduced dsDNA unwinding and G4DNA unfolding, they still retain some activity. It appears that this residual activity is sufficient for mitochondrial function and GCR suppression, but more robust activity is required to suppress the lethality of dna2Δ. We have clarified this interpretation on p11, lines 279-282.

4) While the authors present both DNA binding and unwinding data for the R324 mutations, no data testing whether possible differences in ATPase activity can explain the behavior of R324A/E mutants. Please include ATPase data for the mutants used.

We have included the ATPase data in Supplementary Figure 7e.

Reviewer #2 (Remarks to the Author):

Summary:

In this manuscript, Hong et al. reported their structure-function analysis work on the X-Ray crystal structure of the yeast (*S.cerevisiae*) Pif1 helicase bound to a DNA G-quadruplex (G4). The 3.5 Å structure provided insights into how scPif1 interacts with a DNA G4 molecule and the authors' interpretation of the protein-nucleic acid and protein-protein interactions are tested in biochemical and yeast cellular assays. The authors compared their structure with existing similar structures of Pif1 from other species and deduced that the scPif1 engages G4 using a different mechanism via its wedge region, instead of the 1/2B domains utilized by the ToPif1. The authors focused on an arginine residue (R324) that they identified from their structure and tested its variants' impact on dsDNA and G4 unwinding using in vitro biochemical assays. They found R324 variations had an impact on the helicase unwinding of dsDNA and G4. Further testing of the variants' impact on Pif1's functions in yeast cells yielded a range of effects; R324 is important for Okazaki fragment processing but not for Pif1's roles in respiratory proficiency or suppression of chromosomal rearrangements.

While the X-ray crystal structure of scPif1-G4 and the ensuing biochemical/cellular data are highly informative and interesting, I have several concerns that I believe the authors should address before this body of work is ready for publication. For details, please see the following comments:

Thanks for your detailed comments.

Comments:

1. Page 5, line 93-94: The authors stated that the CD spectrum of scPIF-G4 and the G4 are similar. But Fig. S1b is not showing the two spectra are the same. It will be extremely helpful and less confusing to

mention how the authors interpret the CD spectra and deduce that the AT11 G4 is still intact (i.e, a negative 240 nM band and a positive 260 nM peak are characteristics of a parallel G4 CD spectrum).

Thank you for this suggestion. The AT11 spectrum has a minimum at 240 nm and a maximum at 260 nm, indicative of a parallel G4. When Pif1 is added to AT11, the maximum at 260 nm remains and a minimum at about 230 nm appears that is likely due to Pif1 as α -helices have a minimum in this range. The retention of the peak at 260 nm suggests that binding of Pif1 does not cause unfolding of the AT11 G4 structure. This description has been added to p5, lines 100-101.

2. Page 5, line 112: I highly recommend adding a zoom-in inset for Fig S3 to help the readers see the 2B domain differences between the two structures. In the current version, the figure is cluttered and difficult for readers to find the 2B domain annotation.

Thanks for your suggestions. We have included a zoom-in inset and colored 2B domain in cyan in the structure of ScPif1-ssDNA to highlight the 2B domain differences between the two structures.

3. Page 5, line 116: Figure 1b will benefit from adding more figure panels illustrating different views of the said interactions and include zoom-in insets of various key interactions, especially that of R324. The authors can provide dashed yellow lines connecting two interacting residues to provide the premises to their interaction analysis.

This figure just shows the overall structure of ScPif1-G4 rather than the detailed interactions between ScPif1 and G4. Anyhow, we have included a zoom-in inset to show the wedge region bound to the ssDNA-G4 junction. The key R324 residue is shown as a stick model. The detailed interactions between ScPif1 and G4 are shown in Figure 2.

4. Page 6, line 129-131: the FP binding assay setup is in the titration regime; the probe concentration (10 nM) is similar to the measured K_d values (15-40 nM). Fitting the data with the Hill equation is not adequate in this condition. The quadratic binding equation will be more appropriate. Furthermore, it is unclear if the reactions have reached binding equilibrium (incubation time is not mentioned). I highly encourage the authors to follow the recommendations from Jarmoskaite et al., 2020 eLife to establish the appropriate experimental setup and analysis to measure binding affinities. The R324 variants' binding affinities play an important part in shaping the authors' conclusions.

We agree with your points that fitting the data with the Hill equation is not adequate when the probe concentration (10 nM) is similar to the measured K_d values (15-40 nM). As such, we have reperfomed the FP binding assays using a fluorescence-labeled DNA substrate containing G4 only (AT11-FAM) and fitted the data using the quadratic binding equation. The incubation time for the FP binding assay is added to the method section: Fluorescence anisotropy binding assay (Page 20).

5. Page 6, line 136-138: "These observations strikingly differ from those reported..". A figure showing

this structural difference mentioned by the authors will be helpful for readers' understanding. And why is this difference important?

The 2B domain that the G4 interacts with in the ToPif1 structure is not conserved across Pif1 family helicases making the interaction observed in ToPif1 specific to that enzyme. We have explained the importance of this difference on p6, lines 140-143.

6. Page 6-7, line 144-145: "The stabilized wedge by the Pif1 signature motif would maintain its rigidity for exerting its helicase activity." It is unclear how a rigid wedge can support the Pif1 helicase activity. Please elaborate on this point, preferably with a supplementary figure. This will be extremely helpful for readers not in this area of research.

The two strands of duplex DNA are separated by the wedge, making a rigid wedge critical for helicase activity. We have clarified this on lines 155-156 (p7).

7. Page 7, line 154-155: "The striking difference between ScPif-G4 and ToPif1-G4 is the G4 orientation". Perhaps, the authors can elaborate on if the differences in the G4 DNA sequences used in the two structures would impact the way the two helicases recognize a G4 structure.

Pif1 helicases bind both parallel and anti-parallel G4 DNAs but without sequence specificity. The c-myc G4 used in the ToPif1-G4 structure contains both 3' and 5'tails, each of which bound to one ToPif1 molecule such that the G4 DNA is sandwiched between two ToPif1 molecules. The ToPif1 bound to the 3' tail could impact the orientation of the G4 DNA with respect to the 5' tail. In our ScPif-G4 structure, one ScPif1 bound to the 5' tail mainly contacts the ssDNA-G4 junction. As such, the striking difference in the G4 orientation is not caused by the differences in the G4 DNA sequences used in the two structures and might be caused by the second molecule bound to the 3' tail in the ToPif1-G4 structure.

8. Page 7-8, line 167-177: This section/paragraph is confusing. The section title is the wedge region is a conserved feature. The focus is on R324 of scPif1. The authors described similar residue was found in several species' Pif1 sequences but not for some. So, do they think the wedge is still conserved? If R324 is not conserved in ToPif1, would the earlier comparison of scPif1 and ToPif1 suggest they ToPif1 uses a different mechanism to engage G4?

Thank you for pointing out this lack of clarity. We have on p8, line 188-189 added the explanation that the benefit of a rigid wedge region is due to the importance of the wedge to strand separation during duplex unwinding. Although R324 is not conserved in ToPif1, the general structure and position of the wedge are conserved across the Pif1 family. We agree with the reviewer that ToPif1 uses a different mechanism to engage the G4 that involves the 2B domain as illustrated in Figure 2.

9. Page 8, line 181: Fig S6 is not helpful in seeing what the authors said in the main text. I recommend the authors provide a few more panels and maybe change the colors to better help the readers compare the two aligned structures.

We have regenerated Fig. S6 by providing two different views of the figures and changing the RecD2 colour. In addition, we have clarified the text on p8, line 202-203 and added to the legend for Figure S6 that the wedge is shown in green so that the reader does not have to remember the colour of this region to understand the comparison.

10. Page 8, line 189-191: "...ScPif1 variants that retain the ability to interact with the G-tetrad (R324W and R324Y)...". How do the authors know that these variants still engage the G-tetrad as shown in the WT structure? If they have further structural data supporting these claims, please provide them. If not, the biochemical assays are only saying these variants still bind to the DNA molecule.

We have clarified the text on p9, line 212-213 to indicate that we do not have structural evidence for this interaction being retained, but the interaction is likely to be retained because of the ability of W and Y residues to stack with DNA bases. In addition, the R324W and R324Y variants retain similar binding affinities to the wild type protein (Supplementary Fig. 1e), exhibit high activity (Figure 5) and are fully functional for all tested in vivo activities (Figure 6). This suggests that the structure and critical interactions with the DNA are preserved.

11. Page 8, line 191-193: "...R324W and R324Y ScPif1, which would be predicted to stack with G12 in the 5' most G-tetrad..". It will be extremely helpful for the readers if the authors elaborate why the tryptophan and tyrosine variants will stack with the G12. Are there reported precedents in similar structures? If yes, please include them in the main text.

Our prediction for R324W and R324Y to stack with G12 is based on the interaction of R324 with G12 through cation- π interaction and with T8 combined with the knowledge that aromatic residues frequently interact with nucleotide bases through stacking interactions. This combined with our data showing that substitution of R324 with residues which would not be predicted to interact with DNA bases (R324A and R324E) reduces DNA unwinding and G4DNA unfolding suggests that the R324W and R324Y likely retain the ability to interact with the DNA. We have added references to p9, line 213.

12. Page 9, line 208-209: Please provide quantification of these rates.

Thank you for pointing out this omission. We have added the rate constants to the legend for Figure 5.

13. The cellular assay results are interesting. Perhaps, it will be helpful if the authors can elaborate why they performed these assays, which will be helpful for the readers to understand the motivation tied to the R324 premise. For example, is the DNA-binding or helicase activity of Pif1 critical for these functions? If yes, how would the authors explain the non-effective impact of the tested variants on mitochondria respiratory function and GCR?

The mechanisms by which Pif1 functions in each of these cellular assays are not known so the necessity of DNA binding and helicase activity of Pif1 for these activities are not defined. Because the R324 variants had altered helicase activity but the substitutions did not eliminate helicase activity, we performed these assays to gain insight into the function of Pif1 in these processes. This explanation has been added to p10, line 241-243.

14. Page 11, section “ScPif1 unfolds G4 and unwinds dsDNA via a similar mechanism”: This section should be under the discussion section.

Thanks for your suggestion. The section “ScPif1 unfolds G4 and unwinds dsDNA via a similar mechanism” has been moved to the discussion section.

Minor comments:

1. Page 3, line 56: Please provide the reference(s) for the said smFRET analysis. It is unclear if this information belongs to the reference from the prior sentence.

We have added the reference from the previous sentence to the sentence about smFRET (now line 59).

2. Page 4, line 70-73: I am confused to why the ToPif1 2B domain-G4 interaction provided little information. Maybe the ToPif1 uses a different mechanism from the wedge region that was proposed to be used by other species? There is no introduction to the wedge region before this sentence. Readers will benefit from a sentence or two introducing the wedge region (with references).

We have added a description of the role of the wedge in separating the strands of duplex DNA to p4, line 72-73 along with references. Because this region does not contact the G4 in the ToPif1 structure, it provides little information about the mechanism by which ToPif1 unfolds G4.

3. Page 6, line 124: Figure 2a is not very helpful for the readers to follow the description in the main text. I recommend adding a few more zoom-in insets detailing the interactions that the authors described in the main text.

To help the readers follow the description easier, we have simplified Fig. 2a and included an additional figure with different view (180 degree rotation with respect to the left panel).

4. Page 6, line 144: Fig S5 will benefit from adding interaction details; show contacts by dashed yellow lines between interacting residues.

We have added dashed yellow lines to indicate the contacts between interacting residues.

5. Page 7, line 146-148: A 15 aa swap to alanine is an arguably disruptive mutagenesis approach. It is unclear if the wedge region has become more flexible due to the mutations, or the mutations have perturbed more beyond the flexibility of the wedge. Can the authors elaborate more on how the cited work supports their interpretation?

We agree with the reviewer that this is a disruptive approach. However, this is a common mutational approach to replace a defined structural region (the signature motif) with a flexible linker (alanine residues) to change the structure and eliminate any specific interactions that are occurring. Because this

is disruptive, we have added additional information about single point mutations in the signature sequence of Pif1 to support the role of the signature sequence in supporting the wedge to p7, lines 160-167.

6. Page 7, line 164-164: Please add the references to either the manuscript figures or published work.

We have added a citation and a reference to figure 3 (now page 8, lines 183-185).

7. Page 10, line 227-230: This sentence is too long and confusing. Maybe break into two-three sentences?

We have broken this sentence into two sentences and rephrased it slightly to improve the clarity. (Now p11, lines 252-255.)

8. Page 12, line 279-80: The authors may be intending to use a comma instead of a full stop after “(Fig. 7a)”.

This has been changed to a comma (p13, line 309).

9. Page 16, line 370: “In conclusion, our studies have identified a unified mechanism by which ScPif1 uses a 371 conserved wedge region to unwind dsDNA and to unfold G4 DNA.” I think this a unified mechanism is somewhat overstating given this statement is supported by a single structure from a single species.

We agree with the reviewer that a unified mechanism is somewhat overstating. We replace “ a unified mechanism” with “ a conserved wedge” in the text.

10. No visible error bars for certain datasets in Figure 5e, f. For example, the trap data points do not seems to have error bars plotted in Figure 5e.

All of the data sets in Figure 5e-f have error bars, but they are difficult to see for some data sets (the trap in 5e) because the errors are very small. We have reduced the size of the data points to make the error bars more visible.

11. Page 4, line 92, please define “antiproliferative DNA sequence”.

We have defined anti-proliferative as inhibiting cell growth on p4-5, line 93-94.

12. Most published PDBs do not have their accession numbers cited in their corresponding figure legends or text.

Thank you for pointing out this. We have added the PDB codes for all the published structures in the figure legends.

13. Describe how the authors perform sequence alignment to arrive to their conclusions.

Sequence alignments were performed with Clustal Omega, a tool provided by EMBL at <https://www.ebi.ac.uk/jdispatcher/msa/clustalo>. The legend for Figure S4 now specifies that the alignment is a Clustal Omega multiple sequence alignment.

14. Figure 5b graph is not making use of the space above. I think it will be easier for readers to appreciate the differences (and P values) if they constrain the Y-axis range to between [0, 0.25].

We have changed the y-axis range so that the maximum is 0.25 to make better use of space.

Reviewer #3 (Remarks to the Author):

The Article from Hong and colleagues with the title “ Eukaryotic Pf1 helicase unwinds G-quadruplex and dsDNA using a unified mechanism” is written well however lacks details which makes it hard to follow the conclusions.

In this article they show a co-crystal of *S.cerevisiae* Pif1 DNA helicase with a bound G4 structure. They defined in biochemical experiments the interaction domains of Pif1 with the G4 itself, via the wedge motif. As Pif1 is a multifunctional helicase they characterized further this interaction for Pif1 function in living cells. Further they defined which AS is essential for Okazaki fragment processing. This AS is not important for Pif1 function in the mitochondria or gross chromosomal rearrangements.

Thank you for your suggestions.

Major comments

- Why not a natural G4 from *S. cerevisiae* was used as this would far be more realistic and informative, also more data on how they identified AT11 (screen? Nature) and what AT11 is, is required (sequence, composition)?

We have used a natural G4 from *S. cerevisiae* and other G4s such as c-myc G4, human telomere G4 but none of these G4s stably bind to scPif1 without unfolding. We identified AT11 to form a stable complex with ScPif1 using 1D NMR. AT11 is a modified anti-proliferative DNA aptamer called AS1411. The sequence is provided in Supplementary Table 1.

- Figure 1a, why truncated ScPif1 and not full length? What is the truncation and why is this needed? The truncated version is not introduced. Same A1F4- is not introduced for Figure 1. Is Pif1 truncated in this crystal?

We used the helicase domain of ScPif1 instead of the full length protein because both the N and C-terminal domains are predicted to contain disordered regions, making them unsuitable for crystallography. We have clarified this in the protein purification and crystallization section of the methods (p18). AlF_4^- is aluminium fluoride. We have subscripted the 4 to clarify this. Yes, the truncated Pif1 was used for crystallography. This has been clarified on p19, line 454.

- They used in Figure 1 a 5' tailed G4, how long is the tail and why not a G4 without a tail as published previously was used.

The G4 contains a 7T 5' tail. We have clarified this in the results section (p4, line 106). In addition, the sequence is listed in the methods in the "The G4 DNA substrate T7-AT11 preparation for crystallization" section. Pif1 like other G4 resolving helicases use either 5' or 3' tail to perform translocation driven by ATP hydrolysis. All the previously published structures of DNA helicases bound to a G4 containing a tail.

- Also the complex description is very short and lacks details. Is Pif1 binding as a homodimer? As they state in line 99 : ' the symmetric unit of the crystal structure contains two essentially identical ScPif1 G4 complexes....' What does a rotation of 2B by 90 means?

A single monomer of Pif1 binds to each 7T-AT11 DNA. The crystal unit contains two essentially identical complexes of Pif1-DNA. Rotation of the 2B domain upon G4 binding means that the 2B domain of Pif1 is rotated in the structure of Pif1 bound to G4DNA relative to the 2B domain of Pif1 bound to ssDNA with the remaining domains of Pif1 in essentially the same position in both structures. We have clarified this on p5, line 120.

- They nicely defined the R324 and R326, and K686 and p687 as interacting domains to the G4 of AT11. As Pif1 has different unwinding ability for parallel and anti-parallel is the binding different? How specific is the interaction for this AT11? Here more details on different binding of Pif1 to parallel vs antiparallel is needed. Further which G4 was used in the ToPif1 study, which topology had this G4? Is therefore the interaction different (see line 160 p7).

This structure of ScPif1 can accommodate both parallel and anti-parallel G4s with the same binding mode (Supplemental Figure 9). It can also accommodate a parallel c-Myc G4 in addition to the parallel AT11 G4. Therefore, this interaction is likely conserved for Pif1 interaction with multiple different G4 structures. The structures of ScPif1 and ToPif1 were both solved with parallel G4s so this cannot explain the difference in their interactions with G4. This has been clarified on p7, line 174.

- P6 lines 162 following, information on DHX36 are missing, what is the DSM. Is this helicase truncated? Which G4 structure, topology, conformation? As DHX36 is a 3'-5' and Pif1 a 5'-3' how easy can this be compared? As DHX36 unfold DNA and RNA helicase it would be better to compare BLM or WRN at G4s. Therefore, the conclusion, that Pif1 unfolds differently is OK, but this is expected based on the function and target choice of DHX36 vs Pif1. The strong statement of the authors is too strong and is not holding. Is this true also for other G4 structures? Is Pif1 unique in comparison to other G4 unfolding helicases as it is so far the most efficient G4 unfolding helicase? Here more work is essential.

The DSM is the DHX36 specific motif, a sequence specific to DHX36 that interacts with parallel G4 structures (p3, line 56). Yes, the structure of DHX36-G4 was solved with a truncated version of DHX36 lacking a N-terminal domain. Like the ScPif1-G4 structure, it contains the entire helicase domain. DHX36 binds selectively to parallel G4 structures and was solved with a 3'-ssDNA tailed c-Myc G4 structure. Because helicases of the families containing DHX36 and Pif1 have a channel that binds ssDNA along which they translocate in either the 5'-to-3' or the 3'-to-5' direction by a similar mechanism that involves closing and opening of the 1A and 2A domains upon ATP binding and hydrolysis, the different directions of translocation does not hinder comparison of the structures. Although DHX36 can unfold both G4DNA and G4RNA, it likely interacts with both structures similarly, and the structure was solved bound to G4DNA. We chose to compare the structure of ScPif1-G4 to that of DHX36-G4 and ToPif1-G4 because these are the only structures available of helicases bound to G4s. The description on p3, lines 53-55 has been modified to clarify this.

- Further details are missing on the G4 used in the reporter assay? Does this differ with different G4 substrates? What is with RNA/DNA hybrids?

The full sequence of the G4 substrate used in the reporter assay is listed in Supplementary Table 1, and a schematic diagram of the substrate is shown in Supplementary Figure 8a. It is the same as the c-Myc substrate used in Figure 5 except that it includes a 12 base pair duplex 3' to the G4 to serve as a reporter for G4DNA unfolding. We have added a description of it to the methods (p21, line 514-516). Because the tested variants have similar activity to WT Pif1 and the G4 reporter assay has been previously published using a DNA:DNA duplex, we do not see an advantage to using a RNA:DNA hybrid duplex.

- Further also the *in vivo* experiments are not written detailed enough to understand the experimental setup. Under which promoter is Pif1 expressed in this assay? Where are the Western confirming the expression? Same is true for the mitochondria experiments, is Pif1-from the plasmid expressed in the mitochondria? Or is the truncated version not entering the mitochondria? Same, the GCR rates are done well, however what kind of GCR event was obtained, here Southern blots are needed. And why where no experiments done to test G4 function in cells with the Pif1 variants? (Pif1 DNA binding? GCR at G4s or G4 levels in the cell?)

Thank you for pointing out this omission. We have added a new section to the methods describing the yeast strains and plasmids (p22, lines 526-536). Full length ScPif1 is expressed under control of the *PIF1* promoter in strains lacking endogenous Pif1 so that only the Pif1 variants are expressed. We have added a western blot showing the expression (Supplementary Figure 9). Because Pif1 is expressed from its endogenous promoter and each of the Pif1 R324 variants is active in both the nucleus (Figure 6d) and mitochondria (Figure 6c), we know that the variants are expressed and enter both the nucleus and mitochondria.

The rationale behind the *in vivo* experiments was to determine the effect of Pif1 R324 variants on the ability of Pif1 to perform its functions in cells. One of these is suppression of GCRs. All of the strains expressing R324 variants exhibited low GCR levels indicating they were able to suppress GCRs. Because the GCRs were observed in strains expressing an empty vector and ATPase deficient Pif1 only, and these

have already been characterized elsewhere (PMID: 8287473 and PMID: 11429610), we do not see a benefit to further characterizing these GCR events.

We agree with the reviewer that testing Pif1 function at G4s in cells would be very exciting. However, each Pif1 R324 variant has similar effects on duplex DNA unwinding and G4DNA unfolding, making these variants ill-suited for testing Pif1 G4 function in cells.

REVIEWERS' COMMENTS

Reviewer #1 (Remarks to the Author):

In this revision the authors address the questions that were originally raised, and as a result the current version of the manuscript is significantly improved. No further question or change is required at this time.

Reviewer #2 (Remarks to the Author):

The authors have done well to resolve most, if not all, of my concerns. I have no further comments.

Reviewer #3 (Remarks to the Author):

I would like to thank the authors for addressing all my comments and that they modified their manuscript according to my suggestions